# Structured Prediction
# with Stronger Consistency Guarantees

**Anqi Mao**
Courant Institute
New York, NY 10012
aqmao@cims.nyu.edu

**Mehryar Mohri**
Google Research & CIMS
New York, NY 10011
mohri@google.com

**Yutao Zhong**
Courant Institute
New York, NY 10012
yutao@cims.nyu.edu

## Abstract

We present an extensive study of surrogate losses for structured prediction supported by $\mathcal{H}$-*consistency bounds*. These are recently introduced guarantees that are more relevant to learning than Bayes-consistency, since they are not asymptotic and since they take into account the hypothesis set $\mathcal{H}$ used. We first show that no non-trivial $\mathcal{H}$-consistency bound can be derived for widely used surrogate structured prediction losses. We then define several new families of surrogate losses, including *structured comp-sum losses* and *structured constrained losses*, for which we prove $\mathcal{H}$-consistency bounds and thus Bayes-consistency. These loss functions readily lead to new structured prediction algorithms with stronger theoretical guarantees, based on their minimization. We describe efficient algorithms for minimizing several of these surrogate losses, including a new *structured logistic loss*.

## 1 Introduction

In most applications, the output labels of learning problems have some structure that is crucial to consider. This includes natural language processing applications, where the output may be a sentence, a sequence of parts-of-speech tags, a parse tree, or a dependency graph. It also includes image annotation, image segmentation, computer vision, video annotation, object recognition, motion estimation, computational photography, bioinformatics, and many other important applications.

Several algorithms have been designed in the past for structured prediction tasks, including Conditional Random Fields (CRFs) [Lafferty et al., 2001a, Gimpel and Smith, 2010], StructSVMs [Tsochantaridis et al., 2005a], Maximum-Margin Markov Networks (M3N) [Taskar et al., 2003a], kernel-regression-based algorithms [Cortes et al., 2005, 2007], Voted CRF and StructBoost [Cortes et al., 2016], search-based methods [Daumé III et al., 2009, Doppa et al., 2014, Lam et al., 2015, Chang et al., 2015, Ross et al., 2011] and a variety of deep learning techniques [Jurafsky and Martin, 2009, Vinyals et al., 2015a, Nadeau and Sekine, 2007, Zhang et al., 2008, Wu et al., 2016, Lucchi et al., 2013, Vinyals et al., 2015b], see Appendix A for a more comprehensive list of references and discussion.

Structured prediction tasks inherently involve a natural loss function based on substructures, which could be the Hamming loss, the $n$-gram loss, the edit-distance loss, or some other sequence similarity-based loss or task-specific structured loss. Many of the algorithms previously mentioned overlook this inherent structured loss by simply minimizing the cross-entropy loss. In contrast, the surrogate loss functions minimized by algorithms such as CRFs [Lafferty et al., 2001a, Gimpel and Smith, 2010], M3N [Taskar et al., 2003a], StructSVMs [Tsochantaridis et al., 2005a] or Voted CRF and StructBoost [Cortes et al., 2016] do take into account the natural structured loss of the task. But are these structured prediction loss functions consistent? What guarantees can we rely on when minimizing them over a restricted hypothesis set that does not include all measurable functions? Can we derive non-asymptotic guarantees?

37th Conference on Neural Information Processing Systems (NeurIPS 2023).

This paper deals precisely with these theoretical problems in structured prediction.

**Previous work.** We include a detailed discussion on consistency in structured prediction in Appendix A. Here, we briefly discuss previous work by Osokin et al. [2017]. To our knowledge, this is one of the only studies proving Bayes-consistency for a family of loss functions in structured prediction (see also [Nowak et al., 2020] and other related references in Appendix A). The surrogate losses the authors proposed are the following *quadratic losses* (see also [Zhang, 2004]) defined for any function $h$ mapping $\mathcal{X} \times \mathcal{Y}$ to $\mathbb{R}$ and any loss function $\ell$ between output labels by

$$\forall (x,y) \in \mathcal{X} \times \mathcal{Y}, \quad \mathsf{L}^{\mathrm{quad}}(h,x,y) = \sum_{y' \in \mathcal{Y}} \left[ \ell(y',y) + h(x,y') \right]^2. \tag{1}$$

However, the authors only consider the hypothesis set of linear scoring functions. Moreover, the feature vector in their setting only depends on the input $x$ and ignores the label $y$. In many applications such as natural language prediction, however, it is critical to allow for features that depend both on the input sequence and the output sequence, parse tree, or dependency graph. Finally, in this formulation, the structured prediction problem is cast as a regression problem. Thus, as shown below, the loss function derived is non-standard, even in the binary classification case, where $\ell = \ell_{0-1}$ is the zero-one loss and $\mathcal{Y} = \{y_1, y_2\}$. In this simple case, $\mathsf{L}^{\mathrm{quad}}(h,x,y_1)$ can be expressed as

$$\mathsf{L}^{\mathrm{quad}}(h,x,y_1) = \sum_{y' \in \mathcal{Y}} \left[ \ell_{0-1}(y',y_1) + h(x,y') \right]^2 = h(x,y_1)^2 + (1 + h(x,y_2))^2. \tag{2}$$

This is not a typical formulation since it incorporates the magnitude of individual scores. In contrast, in standard binary classification scenario, only the difference between scores matters.

**Structure of the paper.** We present an extensive study of surrogate losses for structured prediction supported by $\mathcal{H}$-*consistency bounds*. These are recently introduced guarantees that are more relevant to learning than Bayes-consistency, since they are not asymptotic and since they take into account the hypothesis set $\mathcal{H}$ used. We first show that no non-trivial $\mathcal{H}$-consistency bound or even Bayes-consistency can be derived for widely used surrogate structured prediction losses (Section 3). We then define several new families of surrogate losses, including *structured comp-sum losses* (Section 4) and *structured constrained losses* (Section 5), for which we prove $\mathcal{H}$-consistency bounds and thus Bayes-consistency. These loss functions readily lead to new structured prediction algorithms with stronger theoretical guarantees, based on their minimization. We also describe efficient gradient computation algorithms for several of these surrogate losses, including a new *structured logistic loss* (Section 6).

## 2 Preliminaries

**Learning scenario.** We consider the standard structured prediction scenario with the input space $\mathcal{X}$ and output space $\mathcal{Y} = \{1, \ldots, n\}$. The output space may be discrete objects with overlapping structures, such as sequences, images, graphs, parse trees, lists, or others. We assume that the output can be decomposed into $l$ substructures. The substructures could represent words or tokens for example, or other subsequences along a sequence, resulting in the decomposition of the output space $\mathcal{Y}$ as $\mathcal{Y} = \mathcal{Y}_1 \times \cdots \times \mathcal{Y}_l$. Here, each $\mathcal{Y}_j$ represents the set of possible labels or classes that can be assigned to the $j$-th substructure.

**Scoring function.** Structured prediction is typically formulated via *scoring functions* that map $\mathcal{X} \times \mathcal{Y}$ to $\mathbb{R}$, which assign a score to each possible class $y \in \mathcal{Y}$. Let $\mathcal{H}$ be a family of such scoring functions. For any $h \in \mathcal{H}$, we denote by $\mathsf{h}(x)$ its prediction for the input $x \in \mathcal{X}$, which is the output $y \in \mathcal{Y}$ that maximizes the score $h(x,y)$, that is, $\mathsf{h}(x) = \mathrm{argmax}_{y \in \mathcal{Y}} h(x,y)$, with a fixed deterministic strategy to break ties in selecting the label with the highest score. For simplicity, we choose the label with the highest index under the natural ordering of labels as the tie-breaking strategy. We denote by $\mathcal{H}_{\mathrm{all}}$ the family of all measurable scoring functions.

**Generalization error and target loss.** Given a distribution $\mathcal{D}$ over $\mathcal{X} \times \mathcal{Y}$ and a loss function $\mathsf{L} \colon \mathcal{H} \times \mathcal{X} \times \mathcal{Y} \to \mathbb{R}$, the *generalization error* of a hypothesis $h \in \mathcal{H}$ and the *best-in-class generalization error* are defined as follows:

$$\mathcal{R}_{\mathsf{L}}(h) = \mathop{\mathbb{E}}_{(x,y) \sim \mathcal{D}} \left[ \mathsf{L}(h,x,y) \right] \quad \text{and} \quad \mathcal{R}^*_{\mathsf{L},\mathcal{H}} = \inf_{h \in \mathcal{H}} \mathcal{R}_{\mathsf{L}}(h).$$

In structured prediction, the goal is to select a hypothesis $h \in \mathcal{H}$ with small generalization error with respect to a target loss function $\mathsf{L}(h, x, y)$, which can be written as $\mathsf{L}(h, x, y) = \ell(\mathsf{h}(x), y)$ for some non-negative auxiliary function $\ell(y', y)$ with any $y', y \in \mathcal{Y}$. $\ell$ is assumed to be symmetric, that is, $\ell(y', y) = \ell(y, y')$. This is a natural assumption since all instances of $\ell$ that we are familiar with in structured prediction admit this property. A significant characteristic of structured prediction is that the target loss function can be decomposed along the substructures $\mathcal{Y}_k$. As an example, we may use the Hamming loss as the target loss function, which is defined as $\mathsf{L}_{\mathrm{ham}}(h, x, y) = \ell_{\mathrm{ham}}(\mathsf{h}(x), y)$, where $\ell_{\mathrm{ham}}(y', y) = \frac{1}{l} \sum_{j=1}^{l} \mathbb{1}_{y'_j \neq y_j}$ with any $y'_j, y_j \in \mathcal{Y}_j$; we can also use the zero-one loss as the target loss function, which is defined as $\mathsf{L}_{0-1}(h, x, y) = \ell_{0-1}(\mathsf{h}(x), y)$, where $\ell_{0-1}(y', y) = \mathbb{1}_{y' \neq y}$. Note that $\mathsf{L}_{0-1}$ can be viewed as a special case of $\mathsf{L}_{\mathrm{ham}}$ when $l = 1$. We denote by $\ell_{\max} = \max_{y', y \in \mathcal{Y}} \ell(y', y)$ the maximal value of a target loss function. Without loss of generality, we assume that $\ell_{\max} \leq 1$, which can be achieved by normalizing the function $\ell$.

**Consistency guarantees and surrogate loss.** Optimizing the target loss functions in structured prediction for many choices of the hypothesis sets is NP-hard because they are not convex. One common method to address this issue is to resort to surrogate loss functions. Different surrogate loss functions readily lead to different structured prediction algorithms. A natural learning guarantee for such surrogate losses is *Bayes-consistency*, which guarantees that minimizing the generalization error for a surrogate loss $\mathsf{L}_{\mathrm{sur}}$ over $\mathcal{H}_{\mathrm{all}}$ also leads to the minimization of generalization error for the target loss $\mathsf{L}$.

**Definition 1** (Bayes-consistency)**.** *A surrogate loss $\mathsf{L}_{\mathrm{sur}}$ is* Bayes-consistent *in structured prediction, if for any target loss $\ell$, hypothesis $h_n \in \mathcal{H}_{\mathrm{all}}$ and any distribution,*

$$\left( \mathcal{R}_{\mathsf{L}_{\mathrm{sur}}}(h_n) - \mathcal{R}^*_{\mathsf{L}_{\mathrm{sur}}, \mathcal{H}_{\mathrm{all}}} \xrightarrow{n \to +\infty} 0 \right) \implies \left( \mathcal{R}_{\mathsf{L}}(h_n) - \mathcal{R}^*_{\mathsf{L}, \mathcal{H}_{\mathrm{all}}} \xrightarrow{n \to +\infty} 0 \right). \tag{3}$$

Bayes-consistency is an asymptotic guarantee and does not take into account typical hypothesis sets used in structured prediction algorithms, such as linear models or neural networks. To tackle these issues, recent work by Awasthi, Mao, Mohri, and Zhong [2022a,b] propose a stronger consistency guarantee, referred to as $\mathcal{H}$-*consistency bounds*, which are bounds relating the estimation error of the target loss to the estimation error of a surrogate loss (see also [Awasthi et al., 2021a,b, Mao et al., 2023h,e,f, Zheng et al., 2023, Mao et al., 2023b,g,d, Mohri et al., 2023, Mao et al., 2023c,a, Awasthi et al., 2023a,b]):

**Definition 2** ($\mathcal{H}$-consistency bounds)**.** *Given a subset of the hypothesis class $\mathcal{H} \subseteq \mathcal{H}_{\mathrm{all}}$, a surrogate loss $\mathsf{L}_{\mathrm{sur}}$ admits a $\mathcal{H}$-consistency bound in structured prediction, if for some non-decreasing function $f : \mathbb{R}_+ \to \mathbb{R}_+$, a bound of the following form holds for any target loss $\ell$, hypothesis $h \in \mathcal{H}$ and any distribution:*

$$\mathcal{R}_{\mathsf{L}}(h) - \mathcal{R}^*_{\mathsf{L}, \mathcal{H}} \leq f\left( \mathcal{R}_{\mathsf{L}_{\mathrm{sur}}}(h) - \mathcal{R}^*_{\mathsf{L}_{\mathrm{sur}}, \mathcal{H}} \right). \tag{4}$$

As pointed out by Awasthi et al. [2022a,b], $\mathcal{H}$-consistency bounds are the state-of-the-art consistency guarantees for surrogate losses. They are much stronger and more informative than Bayes-consistency, since they account for hypothesis sets $\mathcal{H}$ adopted and provide a quantitative, non-asymptotic relation between surrogate losses and target losses. $\mathcal{H}$-consistency bounds can imply Bayes-consistency when taking $\mathcal{H}$ to be $\mathcal{H}_{\mathrm{all}}$. In the next sections, we will present an extensive study of surrogate losses for structured prediction supported by $\mathcal{H}$-*consistency bounds*.

**Conditional regret and minimizability gap.** We denote by $p(x) = (p(x, 1), \ldots, p(x, n))$ the conditional distribution of $Y$ given $X = x$. Then, the *conditional error* of a hypothesis $h$ for a loss function $\mathsf{L}$, denoted by $\mathcal{C}_{\mathsf{L}}(h, x)$, can be expressed as

$$\mathcal{C}_{\mathsf{L}}(h, x) = \mathop{\mathbb{E}}_{y|x} \left[ \ell(\mathsf{h}(x), y) \right] = \sum_{y \in \mathcal{Y}} p(x, y) \ell(\mathsf{h}(x), y).$$

We further define the best-in-class conditional error and the conditional regret as $\mathcal{C}^*_{\mathsf{L}}(\mathcal{H}, x) = \inf_{h \in \mathcal{H}} \mathcal{C}_{\mathsf{L}}(h, x)$ and $\Delta \mathcal{C}_{\mathsf{L}, \mathcal{H}}(h, x) = \mathcal{C}_{\mathsf{L}}(h, x) - \mathcal{C}^*_{\mathsf{L}}(\mathcal{H}, x)$ respectively. The generalization error can then be rewritten as $\mathcal{R}_{\mathsf{L}}(h) = \mathbb{E}_x[\mathcal{C}_{\mathsf{L}}(h, x)]$.

A key quantity appearing in $\mathcal{H}$-consistency bounds is the *minimizability gap* $\mathcal{M}_{\mathsf{L}}(\mathcal{H})$, which measures the difference between the best-in-class generalization error and the expected best-in-class conditional error for a loss function $\mathsf{L}$ and a hypothesis set $\mathcal{H}$: $\mathcal{M}_{\mathsf{L}}(\mathcal{H}) = \mathcal{R}^*_{\mathsf{L}}(\mathcal{H}) - \mathbb{E}_x[\mathcal{C}^*_{\mathsf{L}}(\mathcal{H}, x)]$. This is an inherent quantity that we cannot hope to minimize or estimate. As shown by Steinwart [2007,

Theorem 3.2], the minimizability gaps vanish $\mathcal{M}_L(\mathcal{H}_{\text{all}}) = 0$ for the family of all measurable functions. More generally, the minimizability gaps vanish when the best-in-class error coincides with the Bayes-error, that is, $\mathcal{R}^*_\ell(\mathcal{H}) = \mathcal{R}^*_\ell(\mathcal{H}_{\text{all}})$ [Awasthi et al., 2022b, Mao et al., 2023h].

The following result characterizes the best-in-class conditional error and the conditional regret for a target loss L, which will be helpful for proving $\mathcal{H}$-consistency bounds in structured prediction. We denote by $H(x)$ the set of all possible predictions on a input $x$ generated by hypotheses in $\mathcal{H}$: $H(x) = \{h(x): h \in \mathcal{H}\}$. The proof is given in Appendix B.

**Lemma 3.** *The best-in-class conditional error and the conditional regret for a target loss* L *in structured prediction can be expressed as follows:*

$$\mathcal{C}^*_{L,\mathcal{H}}(x) = \min_{y' \in H(x)} \sum_{y \in \mathcal{Y}} p(x,y)\ell(y',y)$$

$$\Delta\mathcal{C}_{L,\mathcal{H}}(h,x) = \sum_{y \in \mathcal{Y}} p(x,y)\ell(h(x),y) - \min_{y' \in H(x)} \sum_{y \in \mathcal{Y}} p(x,y)\ell(y',y).$$

## 3 Structured max losses

In this section, we examine the loss functions associated to several prominent structured prediction algorithms. We show that, while they are natural, none of them is Bayes-consistent, which implies that they cannot be supported by $\mathcal{H}$-consistency bounds either. More generally, we consider the following family of surrogate loss functions proposed in [Cortes, Kuznetsov, Mohri, and Yang, 2016], which we refer to as *structured max losses*:

$$\forall (x,y) \in \mathcal{X} \times \mathcal{Y}, \quad L^{\max}(h,x,y) = \max_{y' \neq y} \Phi_{\ell(y',y)}(h(x,y) - h(x,y')), \tag{5}$$

where $\Phi_u: \mathbb{R} \to \mathbb{R}_+$ is an upper bound on $v \mapsto u\mathbb{1}_{v \leq 0}$ for any $u \in \mathbb{R}_+$. In this formulation, different choices of $\Phi_u$ can lead to different structured prediction algorithms. Specifically, as shown by Cortes et al. [2016], the following choices of $\Phi_u(v)$ recover many well-known algorithms:

- $\Phi_u(v) = \max(0, u(1-v))$: *StructSVM* [Tsochantaridis et al., 2005b].

- $\Phi_u(v) = \max(0, u-v)$: *Max-Margin Markov Networks (M3N)* [Taskar et al., 2003b].

- $\Phi_u(v) = \log(1 + e^{u-v})$: *Conditional Random Field (CRF)* [Lafferty et al., 2001b].

- $\Phi_u(v) = ue^{-v}$: *StructBoost* [Cortes et al., 2016].

The following gives a general negative result for $L^{\max}$ that holds under broad assumptions.

**Theorem 4 (Negative results of $L^{\max}$).** *Assume that $n > 2$ and that $\Phi_u(v)$ is convex and non-increasing for $u = 1$. Then, the max structured loss $L^{\max}$ is not Bayes-consistent.*

The proof is included in Appendix C. It is straightforward to see that the assumption of Theorem 4 holds for all the choices of $\Phi_u$ listed above. Thus, the theorem rules out consistency guarantees for any of the loss functions associated to the structured prediction algorithms mentioned above: StructSVM, M3N, CRF, Structboost. Furthermore, Theorem 4 provides negative results for a broad and generalized family of loss functions, collectively referred to as structured max loss. This extends the scope of existing research, as previous works had only addressed the inconsistency of specific instances within the structured max loss category, such as that of M3N [Osokin et al., 2017, Ciliberto et al., 2016, Nowak et al., 2020].

## 4 Structured comp-sum losses

In this section, we first analyze the Voted CRF loss function, which incorporates the auxiliary loss function $\ell$ in the CRF loss and which has been used in several previous studies. Next, we introduce a new family of loss functions for structured predictions that we prove to admit strong consistency guarantees.

### 4.1 Voted Conditional Random Field (VCRF)

We first study a family of surrogate losses called *Voted Conditional Random Field (VCRF)*, which corresponds to the structured prediction algorithm defined in [Cortes et al., 2016]:

$$\forall (x,y) \in \mathcal{X} \times \mathcal{Y}, \mathsf{L}^{\mathrm{VCRF}}(h,x,y) = -\log\left[\frac{e^{h(x,y)}}{\sum_{y' \in \mathcal{Y}} e^{h(x,y')+\ell(y,y')}}\right] = \log\left[\sum_{y' \in \mathcal{Y}} e^{\ell(y,y')+h(x,y')-h(x,y)}\right].$$

This loss function has also been presented as the softmax margin [Gimpel and Smith, 2010] or the reward-augmented maximum likelihood [Norouzi et al., 2016]. It can be viewed as the *softmax variant* of the M3N loss. Indeed, the loss function for M3N can be written as follows:

$$\mathsf{L}(h,x,y) = \max_{y'} \max(0, \ell(y',y) + h(x,y') - h(x,y)). \tag{6}$$

If we replace the maximum function with the softmax, we obtain

$$\mathsf{L}(h,x,y) = \log\left[\sum_{y' \in \mathcal{Y}} e^{\max\left(0,\ell(y',y)+h(x,y')-h(x,y)\right)}\right] = \log\left[\sum_{y' \in \mathcal{Y}} \max\left(1, e^{\ell(y',y)+h(x,y')-h(x,y)}\right)\right]. \tag{7}$$

Next, we show that, as with the loss function for M3N, the VCRF loss function $\mathsf{L}^{\mathrm{VCRF}}$ is inconsistent.

**Theorem 5** (**Negative result of $\mathsf{L}^{\mathrm{VCRF}}$**). *The Voted Conditional Random Field $\mathsf{L}^{\mathrm{VCRF}}$ is not Bayes-consistent.*

The proof is included in Appendix D. The key observation in the proof is that the conditional error of VCRF loss function can be reduced to a specific form when the target loss function $\mathsf{L}$ decouples, which can lead to a different Bayes classifier from that of the target loss function.

To the best of our knowledge, no prior studies in the literature have explored the consistency of the VCRF loss formulation. The most closely related discussions center around a specialized instance of the multi-class logistic loss (also referred to as Conditional Random Field in that context), in which $\ell(y',y)$ disappears within the framework of the Voted Conditional Random Field. The previous works by Osokin et al. [2017], Ciliberto et al. [2016], Nowak et al. [2020] point out that the multi-class logistic loss cannot be consistent in structured prediction due to the absence of the target loss function within its formulation. Instead, our result shows that, even when integrating the target loss $\ell(y',y)$ within its formulation, the Voted Conditional Random Field cannot be consistent.

Along with Theorem 4, these results rule out consistency guarantees for commonly used surrogate loss functions in structured prediction.

### 4.2 Structured comp-sum loss functions

In this section, we define a family of new loss functions for structured prediction that are not only Bayes-consistent but also supported by $\mathcal{H}$-consistency bounds. These are loss functions that can be viewed as the generalization to structured prediction of loss functions defined via a composition and a sum, and that have been referred to as *comp-sum losses* in [Mao et al., 2023h]. Thus, we will refer to them as *structured comp-sum losses*. They are defined as follows:

$$\forall (x,y) \in \mathcal{X} \times \mathcal{Y}, \quad \mathsf{L}^{\mathrm{comp}}(h,x,y) = \sum_{y' \in \mathcal{Y}} \overline{\ell}(y',y)\Phi_1\left(\sum_{y'' \in \mathcal{Y}} \Phi_2(h(x,y'') - h(x,y'))\right), \tag{8}$$

where $\overline{\ell}(y',y) = 1 - \ell(y',y)$, $\Phi_1 \colon \mathbb{R}_+ \to \mathbb{R}_+$ is a non-decreasing auxiliary function and $\Phi_2 \colon \mathbb{R} \to \mathbb{R}_+$ a non-decreasing auxiliary function. This formulation (8) can also be viewed as a weighted comp-sum loss, if we interpret $\overline{\ell}(\cdot,y)$ as a weight vector.

Specifically, we can choose $\Phi_2(v) = e^v$ and $\Phi_1(v) = \log(v)$, $\Phi_1(v) = v-1$, $\Phi_1(v) = \frac{1}{\alpha}\left(1 - \frac{1}{v^\alpha}\right), \alpha \in (0,1)$ and $\Phi_1(v) = 1 - \frac{1}{v}$, which leads to new surrogate losses for structured prediction defined in Table 1. These surrogate losses are novel strict generalization of their counterparts in the standard multi-class classification case where $\ell = \ell_{0-1}$. More precisely, when $\ell = \ell_{0-1}$, $\mathsf{L}^{\mathrm{comp}}_{\log}$ coincides with the *logistic loss* [Verhulst, 1838, 1845, Berkson, 1944, 1951]; $\mathsf{L}^{\mathrm{comp}}_{\exp}$ coincides with the *sum-exponential loss* [Weston and Watkins, 1998, Awasthi et al., 2022b]; $\mathsf{L}^{\mathrm{comp}}_{\mathrm{gce}}$ coincides with the

Table 1: A new family of surrogate losses for structured prediction: structured comp-sum losses.

| $\Phi_1(v)$ | Name | Formulation |
|---|---|---|
| $\log(v)$ | Structured logistic loss | $\mathsf{L}_{\log}^{\mathrm{comp}} = -\sum_{y'\in\mathcal{Y}} \overline{\ell}(y',y) \log\left[\frac{e^{h(x,y')}}{\sum_{y''\in\mathcal{Y}} e^{h(x,y'')}}\right].$ |
| $v-1$ | Structured sum-exponential loss | $\mathsf{L}_{\exp}^{\mathrm{comp}} = \sum_{y'\in\mathcal{Y}} \overline{\ell}(y',y) \sum_{y''\neq y'} e^{h(x,y'')-h(x,y')}$ |
| $\frac{1}{\alpha}\left[1-\frac{1}{v^\alpha}\right]$ | Structured generalized cross-entropy loss | $\mathsf{L}_{\mathrm{gce}}^{\mathrm{comp}} = \sum_{y'\in\mathcal{Y}} \overline{\ell}(y',y)\frac{1}{\alpha}\left[1-\left[\frac{e^{h(x,y')}}{\sum_{y''\in\mathcal{Y}} e^{h(x,y'')}}\right]^\alpha\right]$ |
| $1-\frac{1}{v}$ | Structured mean absolute error loss | $\mathsf{L}_{\mathrm{mae}}^{\mathrm{comp}} = \sum_{y'\in\mathcal{Y}} \overline{\ell}(y',y)\left[1-\frac{e^{h(x,y')}}{\sum_{y''\in\mathcal{Y}} e^{h(x,y'')}}\right].$ |

*generalized cross-entropy loss* [Zhang and Sabuncu, 2018]; and $\mathsf{L}_{\mathrm{mae}}^{\mathrm{comp}}$ coincides with the *mean absolute error loss* [Ghosh et al., 2017].

We will show that these structured comp-sum losses benefit from $\mathcal{H}$-consistency bounds in structured prediction, when $\mathcal{H}$ is a *symmetric* and *complete* hypothesis set. A hypothesis set $\mathcal{H}$ is *symmetric* if there exists a family $\mathcal{F}$ of real-valued functions such that $\{[h(x,1),\ldots,h(x,n)]:h\in\mathcal{H}\} = \{[f_1(x),\ldots,f_n(x)]:f_1,\ldots,f_n\in\mathcal{F}\}$ for any $x\in\mathcal{X}$. Thus, the choice of the scoring functions does not depend on the order of the categories in $\mathcal{Y}$. A hypothesis set $\mathcal{H}$ is *complete* if it can generate scores that span $\mathbb{R}$, that is, $\{h(x,y):h\in\mathcal{H}\} = \mathbb{R}$ for any $(x,y)\in\mathcal{X}\times\mathcal{Y}$. As shown by Awasthi et al. [2022b] and Mao et al. [2023h], these assumptions are general and hold for common hypothesis sets used in practice, such as the family of linear hypotheses and that of multi-layer feed-forward neural networks, and of course that of all measurable functions.

**Theorem 6** ($\mathcal{H}$**-consistency bound of** $\mathsf{L}^{\mathrm{comp}}$). *Assume that $\mathcal{H}$ is symmetric and complete. Then, for any target loss $\ell$, any hypothesis $h\in\mathcal{H}$ and any distribution, we have*

$$\mathcal{R}_{\mathsf{L}}(h) - \mathcal{R}_{\mathsf{L},\mathcal{H}}^* \leq \Gamma\big(\mathcal{R}_{\mathsf{L}^{\mathrm{comp}}}(h) - \mathcal{R}_{\mathsf{L}^{\mathrm{comp}},\mathcal{H}}^* + \mathcal{M}_{\mathsf{L}^{\mathrm{comp}},\mathcal{H}}\big) - \mathcal{M}_{\mathsf{L},\mathcal{H}}, \qquad (9)$$

*where $\Gamma(t) = 2\sqrt{t}$ when $\mathsf{L}^{\mathrm{comp}} = \mathsf{L}_{\log}^{\mathrm{comp}}$ or $\mathsf{L}_{\exp}^{\mathrm{comp}}$; $\Gamma(t) = 2\sqrt{n^\alpha t}$ when $\mathsf{L}^{\mathrm{comp}} = \mathsf{L}_{\mathrm{gce}}^{\mathrm{comp}}$; and $\Gamma(t) = nt$ when $\mathsf{L}^{\mathrm{comp}} = \mathsf{L}_{\mathrm{mae}}^{\mathrm{comp}}$.*

Theorem 6 represents a consolidated result for the four structured comp-sum losses, with the proofs for each being presented separately in Appendix E. The key step of the proof is to upper bound the conditional regret of the target loss (Lemma 3) by that of a surrogate loss. To achieve this, we upper bound the best-in-class conditional error by the conditional error of a carefully selected hypothesis $\overline{h}_\mu\in\overline{\mathcal{H}}$. The resulting softmax $\overline{\mathcal{S}}_\mu$ of this hypothesis only differs from the original softmax $\mathcal{S}$ corresponding to $\overline{h}$ on two labels. Theorem 6 admits as special cases the $\mathcal{H}$-consistency bounds of Mao et al. [2023h] given for standard multi-class classification ($\ell = \ell_{0-1}$) and significantly extends them to the general structured prediction scenario.

Let us emphasize that our proof technique is novel and distinct from the approach used in [Mao et al., 2023h], which only applies to the special case where $\ell$ is the zero-one loss and cannot be generalized to any target loss $\ell$. In their proof, the authors choose $\overline{h}_\mu$ based on individual scores $\overline{h}(x,y)$, rather than the softmax. Consequently, when $\ell\neq\ell_{0-1}$, as is common in structured prediction, the resulting optimization problem of $\mu$ can be very intricate and a closed-form expression of the optimization solution cannot be derived. However, our new proof method overcomes this limitation. By viewing the softmax of hypothesis as a unit and introducing a pseudo-conditional distribution $\overline{q}$, we are able to solve a simple constrained optimization problem on $\mu$ within structured prediction scenario.

By Steinwart [2007, Theorem 3.2], the minimizability gaps $\mathcal{M}_{\mathsf{L}^{\mathrm{comp}},\mathcal{H}}$ and $\mathcal{M}_{\mathsf{L},\mathcal{H}}$ vanish for the family of all measurable functions. Therefore, when $\mathcal{H} = \mathcal{H}_{\mathrm{all}}$, the $\mathcal{H}$-consistency bounds provided in Theorem 6 imply the Bayes-consistency of these structured comp-sum losses.

**Corollary 7.** *The structured comp-sum loss $\mathsf{L}^{\mathrm{comp}}$ is Bayes-consistent for $\mathsf{L}^{\mathrm{comp}} = \mathsf{L}_{\log}^{\mathrm{comp}}$, $\mathsf{L}_{\exp}^{\mathrm{comp}}$, $\mathsf{L}_{\mathrm{gce}}^{\mathrm{comp}}$, and $\mathsf{L}_{\mathrm{mae}}^{\mathrm{comp}}$.*

In fact, Theorem 6 provides stronger quantitative bounds than Bayes-consistency when the minimizability gaps vanish, which suggests that if the estimation error of the structured comp-sum loss $\mathcal{R}_{\mathsf{L}^{\mathrm{comp}}}(h) - \mathcal{R}_{\mathsf{L}^{\mathrm{comp}},\mathcal{H}}^*$ is reduced to $\epsilon$, the estimation error of the target loss $\mathcal{R}_{\mathsf{L}}(h) - \mathcal{R}_{\mathsf{L},\mathcal{H}}^*$ is upper bounded by $2\sqrt{\epsilon}$ for structured logistic loss and structured sum-exponential loss, $2\sqrt{n^\alpha\epsilon}$ for structured generalized cross-entropy loss, and $n\epsilon$ for structured mean absolute error loss.

Table 2: A new family of surrogate losses for structured prediction: structured constrained losses.

| $\Phi_u(v)$ | Name | Formulation ($\sum_{y\in\mathcal{Y}} h(x,y) = 0$) |
|---|---|---|
| $ue^{-v}$ | Structured constrained exponential loss | $\mathsf{L}_{\exp}^{\mathrm{cstnd}} = \sum_{y'\in\mathcal{Y}} \ell(y',y)\max\{0,1-h(x,y')\}^2$ |
| $u\max\{0,1-v\}^2$ | Structured constrained squared-hinge loss | $\mathsf{L}_{\mathrm{hinge}}^{\mathrm{cstnd}} = \sum_{y'\in\mathcal{Y}} \ell(y',y)\max\{0,1-h(x,y')\}$ |
| $u\max\{0,1-v\}$ | Structured constrained hinge loss | $\mathsf{L}_{\mathrm{hinge}}^{\mathrm{cstnd}} = \sum_{y'\in\mathcal{Y}} \ell(y',y)\max\{0,1-h(x,y')\}$ |
| $u\min\{\max\{0,1-\frac{v}{\rho}\},1\}$ | Structured constrained $\rho$-margin loss | $\mathsf{L}_{\rho}^{\mathrm{cstnd}} = \sum_{y'\in\mathcal{Y}} \ell(y',y)\min\{\max\{0,1-\frac{h(x,y')}{\rho}\},1\}.$ |

# 5 Structured constrained loss functions

In this section, we introduce another new family of surrogate losses for structured prediction that we prove to admit $\mathcal{H}$-consistency bounds. We will present a novel generalization of the *constrained losses* [Lee et al., 2004, Awasthi et al., 2022b] to structured prediction. Thus, we refer to them as *structured constrained losses* and define them as follows:

$$\forall (x,y) \in \mathcal{X} \times \mathcal{Y}, \quad \mathsf{L}^{\mathrm{cstnd}}(h,x,y) = \sum_{y'\in\mathcal{Y}} \Phi_{\ell(y',y)}(-h(x,y')), \tag{10}$$

with the constraint that $\sum_{y\in\mathcal{Y}} h(x,y) = 0$ and $\Phi_u\colon\mathbb{R} \to \mathbb{R}_+$ is an upper bound on $v \mapsto u\mathbb{1}_{v\le 0}$ for any $u \in \mathbb{R}_+$. In standard constrained loss formulation, a single-variable function $\Phi(v)$ that defines a margin-based loss is used. In (10), the single-variable function $\Phi(v)$ is generalized to being a function of two variables $\Phi_u(v)$, which depends on both the target loss and the scores, to accommodate the structured prediction scenario. Specifically, we can choose $\Phi_u(v) = ue^{-v}$, $\Phi_u(v) = u\max\{0,1-v\}^2$, $\Phi_u(v) = u\max\{0,1-v\}$, $\Phi_u(v) = u\min\{\max\{0,1-v/\rho\},1\}$, which lead to new surrogate losses for structured prediction defined in Table 2. These surrogate losses are novel generalization of their corresponding counterparts [Lee et al., 2004, Awasthi et al., 2022b] in standard multi-class classification, where $\ell = \ell_{0-1}$. As with structured comp-sum losses, we will show that these structured constrained losses benefit from $\mathcal{H}$-consistency bounds in structured prediction as well, for any symmetric and complete hypothesis set $\mathcal{H}$.

**Theorem 8** ($\mathcal{H}$-**consistency bound of** $\mathsf{L}^{\mathrm{cstnd}}$). *Assume that $\mathcal{H}$ is symmetric and complete. Then, for any target loss $\ell$, hypothesis $h \in \mathcal{H}$ and any distribution, we have*

$$\mathcal{R}_{\mathsf{L}}(h) - \mathcal{R}_{\mathsf{L},\mathcal{H}}^* \le \Gamma\big(\mathcal{R}_{\mathsf{L}^{\mathrm{cstnd}}}(h) - \mathcal{R}_{\mathsf{L}^{\mathrm{cstnd}},\mathcal{H}}^* + \mathcal{M}_{\mathsf{L}^{\mathrm{cstnd}},\mathcal{H}}\big) - \mathcal{M}_{\mathsf{L},\mathcal{H}}. \tag{11}$$

*where $\Gamma(t) = 2\sqrt{\ell_{\max}t}$ when $\mathsf{L}^{\mathrm{cstnd}} = \mathsf{L}_{\exp}^{\mathrm{cstnd}}$; $\Gamma(t) = 2\sqrt{t}$ when $\mathsf{L}^{\mathrm{cstnd}} = \mathsf{L}_{\mathrm{sq-hinge}}^{\mathrm{cstnd}}$; and $\Gamma(t) = t$ when $\mathsf{L}^{\mathrm{cstnd}} = \mathsf{L}_{\mathrm{hinge}}^{\mathrm{cstnd}}$ or $\mathsf{L}_{\rho}^{\mathrm{cstnd}}$.*

The proof is included in Appendix F. As for Theorem 8, the key part of the proof is to upper bound the conditional regret of the target loss (Lemma 3) by that of a surrogate loss. Here too, we introduce a pseudo-conditional distribution $q$, which can be viewed as a weighted distribution of the original one, $p(x)$, with weights given by the target loss function. Then, we upper bound the best-in-class conditional error by the conditional error of a carefully selected hypothesis $\overline{h}_\mu \in \overline{\mathcal{H}}$.

As shown by Steinwart [2007, Theorem 3.2], for the family of all measurable functions, the minimizability gaps vanish: $\mathcal{M}_{\mathsf{L}^{\mathrm{cstnd}},\mathcal{H}} = 0$ and $\mathcal{M}_{\mathsf{L},\mathcal{H}} = 0$. Therefore, when $\mathcal{H} = \mathcal{H}_{\mathrm{all}}$, the $\mathcal{H}$-consistency bounds provided in Theorem 6 imply the Bayes-consistency of these structured constrained losses.

**Corollary 9.** *The structured constrained loss $\mathsf{L}^{\mathrm{cstnd}}$ is Bayes-consistent for $\mathsf{L}^{\mathrm{cstnd}} = \mathsf{L}_{\exp}^{\mathrm{cstnd}}$, $\mathsf{L}_{\mathrm{sq-hinge}}^{\mathrm{cstnd}}$, $\mathsf{L}_{\mathrm{hinge}}^{\mathrm{cstnd}}$, and $\mathsf{L}_{\rho}^{\mathrm{cstnd}}$.*

As with the cases of structured comp-sum losses, Theorem 8 provides in fact stronger quantitative bounds than Bayes-consistency. They show that that if the estimation error of the structured constrained loss $\mathcal{R}_{\mathsf{L}^{\mathrm{comp}}}(h) - \mathcal{R}_{\mathsf{L}^{\mathrm{comp}},\mathcal{H}}^*$ is reduced to $\epsilon$, the estimation error of the target loss $\mathcal{R}_{\mathsf{L}}(h) - \mathcal{R}_{\mathsf{L},\mathcal{H}}^*$ is upper bounded by $2\sqrt{\ell_{\max}\epsilon}$ for $\mathsf{L}_{\exp}^{\mathrm{cstnd}}$, $2\sqrt{\epsilon}$ for $\mathsf{L}_{\mathrm{sq-hinge}}^{\mathrm{cstnd}}$ and $\epsilon$ for $\mathsf{L}_{\mathrm{hinge}}^{\mathrm{cstnd}}$ and $\mathsf{L}_{\rho}^{\mathrm{cstnd}}$.

It is important to note that we can upper bound the minimizability gap by the approximation error, or finer terms depending on the magnitude of the parameter space as in [Mao et al., 2023h]. Furthermore, our $\mathcal{H}$-consistency bounds (Theorems 6 and 8) can be used to derive finite sample learning bounds for a hypothesis set $\mathcal{H}$. These bounds depend on the Rademacher complexity of the hypothesis set and the loss function, as well as an upper bound on the minimizability gap for the surrogate loss.

# 6  Optimization of $\mathsf{L}_{\log}^{\mathrm{comp}}$ and $\mathsf{L}_{\exp}^{\mathrm{comp}}$

In this section, we show that the gradient of the structured logistic loss $\mathsf{L}_{\log}^{\mathrm{comp}}$ can be computed efficiently at any point $(x_i, y_i)$ and therefore that this loss function is both supported by $\mathcal{H}$-consistency bounds and is of practical use. We similarly show that for $\mathsf{L}_{\exp}^{\mathrm{comp}}$ in Appendix G.2.

Fix the labeled pair $(x_i, y_i)$ and $h \in \mathcal{H}$. Observe that $\mathsf{L}_{\log}^{\mathrm{comp}}(h, x_i, y_i)$ can be equivalently rewritten as follows:

$$
\begin{aligned}
\mathsf{L}_{\log}^{\mathrm{comp}}(h, x_i, y_i) &= \sum_{y' \in \mathcal{Y}} \overline{\ell}(y', y_i) \log\left[ \sum_{y'' \in \mathcal{Y}} e^{h(x_i, y'') - h(x_i, y')} \right] \\
&= -\sum_{y' \in \mathcal{Y}} \overline{\ell}(y', y_i) h(x_i, y') + \left[ \sum_{y' \in \mathcal{Y}} \overline{\ell}(y', y_i) \right] \log\left[ \sum_{y'' \in \mathcal{Y}} e^{h(x_i, y'')} \right] \\
&= -\sum_{y' \in \mathcal{Y}} \overline{\ell}(y', y_i) h(x_i, y') + \overline{\ell}_i \log Z_{h,i},
\end{aligned}
$$

where $\overline{\ell}_i = \sum_{y' \in \mathcal{Y}} \overline{\ell}(y', y_i)$, and $Z_{h,i} = \sum_{y \in \mathcal{Y}} e^{h(x_i, y)}$. Note that $\overline{\ell}_i$ does not depend on $h$ and can be pre-computed. Modulo normalization, this quantity is the average *similarity* of $y_i$ to $\mathcal{Y}$, if we interpret $\overline{\ell} = 1 - \ell$ as a similarity. While $\mathcal{Y}$ may be very large, this can be often computed straightforwardly for most loss functions $\ell$. For example, for the Hamming loss, for sequences of length $l$, we have

$$
\frac{1}{|\mathcal{Y}|} \overline{\ell}_i = \frac{1}{l} \mathbb{E}\left[ \sum_{k=1}^{l} \left(1 - \mathbb{1}_{y'_k \neq y_{i,k}} \right) \right] = \frac{1}{l} \sum_{k=1}^{l} \mathbb{E}\left[ \mathbb{1}_{y'_k = y_{i,k}} \right] = \frac{1}{l} \sum_{k=1}^{l} \frac{1}{2} = \frac{1}{2}.
$$

Thus, in this case, $\overline{\ell}_i$ does not depend on $i$ and is a universal constant. Similarly, $\overline{\ell}_i$ can be shown to be a constant for many other losses.

**Hypothesis set.** For the remaining of this section, to simplify the presentation, we will consider the hypothesis set of linear functions $\mathcal{H} = \left\{ x \mapsto \mathbf{w} \cdot \Psi(x, y) \colon \mathbf{w} \in \mathbb{R}^d \right\}$, where $\Psi$ is a feature mapping from $\mathcal{X} \times \mathcal{Y}$ to $\mathbb{R}^d$. Note that a number of classical structured prediction algorithms adopt the same linear hypothesis set: StructSVM [Tsochantaridis et al., 2005b], Max-Margin Markov Networks (M3N) [Taskar et al., 2003b], Conditional Random Field (CRF) [Lafferty et al., 2001b], Voted Conditional Random Field (VCRF) [Cortes et al., 2016]. Our algorithms can also be incorporated into standard procedures for training neural network architectures (see [Cortes et al., 2018], Appendix B).

**Markovian features.** We will also assume Markovian features, as is common in structured prediction. Features used in practice often admit this property. Furthermore, in the absence of any such assumption, it is known that learning and inference in general are intractable. We will largely adopt here the definitions and notation from [Cortes et al., 2016] and will consider the common case where $\mathcal{Y}$ is a set of sequences of length $l$ over a finite alphabet $\Delta$ of size $r$. Other structured problems can be treated in similar ways. We will denote by $\varepsilon$ the empty string and for any sequence $y = (y_1, \ldots, y_l) \in \mathcal{Y}$, we will denote by $y_s^{s'} = (y_s, \ldots, y_{s'})$ the substring of $y$ starting at index $s$ and ending at $s'$. For convenience, for $s \leq 0$, we define $y_s$ by $y_s = \varepsilon$.

We will assume that the feature vector $\Psi$ admits a *Markovian property of order $q$*, that is it can be decomposed as follows for any $(x, y) \in \mathcal{X} \times \mathcal{Y}$:

$$
\Psi(x, y) = \sum_{s=1}^{l} \psi(x, y_{s-q+1}^s, s). \tag{12}
$$

for some position-dependent feature vector function $\psi$ defined over $\mathcal{X} \times \Delta^q \times [l]$. We note that we can write $\Psi = \sum_{k=1}^{p} \tilde{\Psi}_k$ with $\tilde{\Psi}_k = (0, \ldots, \Psi_k, \ldots, 0)$. In the following, abusing the notation, we will simply write $\Psi_k$ instead of $\tilde{\Psi}_k$. Each $\Psi_k$ corresponds to a Markovian feature vector based only on $k$-grams, $p$ is the largest $k$. Thus, for any $x \in \mathcal{X}$ and $y \in \mathcal{Y}$, we have

$$
\Psi(\mathbf{x}, y) = \sum_{k=1}^{p} \Psi_k(x, y). \tag{13}
$$

For any $k \in [1, p]$, let $\psi_k$ denote the position-dependent feature vector function corresponding to $\Psi_k$. Also, for any $x \in \mathcal{X}$ and $y \in \Delta^l$, define $\widetilde{\psi}$ by $\widetilde{\psi}(x, y_{s-p+1}^s, s) = \sum_{k=1}^p \psi_k(x, y_{s-k+1}^s, s)$. Observe then that we can write

$$\Psi(x, y) = \sum_{k=1}^p \Psi_k(x, y) = \sum_{k=1}^p \sum_{s=1}^l \psi_k(x, y_{s-k+1}^s, s) = \sum_{s=1}^l \sum_{k=1}^p \psi_k(x, y_{s-k+1}^s, s) = \sum_{s=1}^l \widetilde{\psi}(x, y_{s-p+1}^s, s).$$

**Gradient computation.** Adopting the shorthand $\mathbf{w}$ for $h$, we can rewrite the loss at $(x_i, y_i)$ as:

$$\mathsf{L}_{\log}^{\mathrm{comp}}(\mathbf{w}, x_i, y_i) = -\mathbf{w} \cdot \left[ \sum_{y' \in \mathcal{Y}} \overline{\ell}(y', y_i) \Psi(x_i, y') \right] + \overline{\ell}_i \log Z_{\mathbf{w}, i}.$$

Thus, the gradient of $\mathsf{L}_{\log}^{\mathrm{comp}}$ at any $\mathbf{w} \in \mathbb{R}^d$ is given by

$$\nabla \mathsf{L}_{\log}^{\mathrm{comp}}(\mathbf{w}) = -\sum_{y' \in \mathcal{Y}} \overline{\ell}(y', y_i) \Psi(x_i, y') + \overline{\ell}_i \sum_{y \in \mathcal{Y}} \frac{e^{\mathbf{w} \cdot \Psi(x_i, y)}}{\sum_{y'' \in \mathcal{Y}} e^{\mathbf{w} \cdot \Psi(x_i, y'')}} \Psi(x_i, y)$$

$$= -\sum_{y' \in \mathcal{Y}} \overline{\ell}(y', y_i) \Psi(x_i, y') + \overline{\ell}_i \mathop{\mathbb{E}}_{y \sim \mathsf{q}_{\mathbf{w}}} [\Psi(x_i, y)],$$

where $\mathsf{q}_{\mathbf{w}}$ is defined for all $y \in \mathcal{Y}$ by $\mathsf{q}_{\mathbf{w}}(y) = \frac{e^{\mathbf{w} \cdot \Psi(x_i, y)}}{Z_{\mathbf{w}}}$ with $Z_{\mathbf{w}} = \sum_{y \in \mathcal{Y}} e^{\mathbf{w} \cdot \Psi(x_i, y)}$. Note that the sum defining these terms is over a number of sequences $y$ that is exponential in $r$ and that the computation appears to be therefore challenging. The following lemma gives the expression of the gradient of $\mathsf{L}_{\log}^{\mathrm{comp}}$ and helps identify the most computationally challenging terms.

**Lemma 10.** *For any $\mathbf{w} \in \mathbb{R}^d$, the gradient of $\mathsf{L}_{\log}^{\mathrm{comp}}$ can be expressed as follows:*

$$\nabla \mathsf{L}_{\log}^{\mathrm{comp}}(\mathbf{w}) = \sum_{s=1}^l \sum_{\mathbf{z} \in \Delta^p} \left[ \overline{\ell}_i \mathsf{Q}_{\mathbf{w}}(\mathbf{z}, s) - \mathsf{L}(\mathbf{z}, s) \right] \widetilde{\psi}(x_i, \mathbf{z}, s),$$

*where $\mathsf{Q}_{\mathbf{w}}(\mathbf{z}, s) = \sum_{y : y_{s-p+1}^s = \mathbf{z}} \mathsf{q}_{\mathbf{w}}(y)$ and $\mathsf{L}(\mathbf{z}, s) = \sum_{y : y_{s-p+1}^s = \mathbf{z}} \overline{\ell}(y, y_i)$.*

*Proof.* Using the decomposition of the feature vector, we can write:

$$\sum_{y \in \mathcal{Y}} \overline{\ell}(y, y_i) \Psi(x_i, y) = \sum_{y \in \Delta^l} \overline{\ell}(y, y_i) \sum_{s=1}^l \widetilde{\psi}(x_i, y_{s-p+1}^s, s) = \sum_{s=1}^l \sum_{\mathbf{z} \in \Delta^p} \left[ \sum_{y : y_{s-p+1}^s = \mathbf{z}} \overline{\ell}(y, y_i) \right] \widetilde{\psi}(x_i, \mathbf{z}, s)$$

$$\mathop{\mathbb{E}}_{y \sim \mathsf{q}_{\mathbf{w}}} [\Psi(x_i, y)] = \sum_{y \in \Delta^l} \mathsf{q}_{\mathbf{w}}(y) \sum_{s=1}^l \widetilde{\psi}(x_i, y_{s-p+1}^s, s) = \sum_{s=1}^l \sum_{\mathbf{z} \in \Delta^p} \left[ \sum_{y : y_{s-p+1}^s = \mathbf{z}} \mathsf{q}_{\mathbf{w}}(y) \right] \widetilde{\psi}(x_i, \mathbf{z}, s).$$

This completes the proof. $\qquad\square$

In light of this result, the bottleneck in the gradient computation is the evaluation of $\mathsf{Q}_{\mathbf{w}}(\mathbf{z}, s)$ and $\mathsf{L}(\mathbf{z}, s)$ for all $s \in [l]$ and $\mathbf{z} \in \Delta^p$. In previous work [Cortes, Kuznetsov, Mohri, and Yang, 2016, Cortes, Kuznetsov, Mohri, Storcheus, and Yang, 2018], it was shown that the quantities $\mathsf{Q}_{\mathbf{w}}(\mathbf{z}, s)$ can be determined efficiently, all together, by running two single-source shortest-distance algorithms over the $(+, \times)$ semiring on an appropriate weighted finite automaton (WFA). The overall time complexity of the computation of all quantities $\mathsf{Q}_{\mathbf{w}}(\mathbf{z}, s)$, $\mathbf{z} \in \Delta^p$ and $s \in [l]$, is then in $O(lr^p)$, where $r = |\Delta|$.

We now analyze the computation of $\mathsf{L}(\mathbf{z}, s)$ for a fixed $\mathbf{z} \in \Delta^p$ and $s \in [l]$. Note that, unlike $\mathsf{Q}_{\mathbf{w}}(\mathbf{z}, s)$, this term does not depend on $\mathbf{w}$ and can therefore be computed once and for all, before any gradient computation. The sum defining $\mathsf{L}(\mathbf{z}, s)$ is over all sequences $y$ that admit the substring $\mathbf{z}$ at position $s$.

**Rational losses.** In Appendix G.1, we also give an efficient algorithm for the computation of the quantities $\mathsf{L}(\mathbf{z}, s)$ in the case of Markovian losses. Here, we present an efficient algorithm for their computation in the important case of *rational losses*. This is a general family of loss functions based on rational kernels [Cortes, Haffner, and Mohri, 2004] that includes, in particular, $n$-gram losses, which can be defined for a pair of sequences $(y, y')$ as the negative inner product of the vectors of $n$-gram counts of $y$ and $y'$.

Our algorithm bears some similarity to that of Cortes et al. [2018] for the computation of the gradient of the VCRF loss function. It is however distinct because the structured prediction loss function we are considering and our definition of rational loss are both different. We will adopt a similar notation and terminology. Recall that for any sequence $y$, we denote by $y_i$ the symbol in its $i$th position and by $y_i^j = y_i y_{i+1} \cdots y_j$ the substring of $y$ starting at position $i$ and ending at $j$. We denote by $\mathsf{E}_{\mathcal{A}}$ the set of transitions of a WFA $\mathcal{A}$. Let $\mathcal{U}$ be a weighted finite-state transducer (WFST) over the $(+, \times)$ semiring over the reals, with $\Delta$ as both the input and output alphabet. Then, we define the rational loss associated to $\mathcal{U}$ for all $y, y' \in \Delta^*$ by $\overline{\ell}(y, y') = \mathcal{U}(y, y')$.

Let $\overline{\mathcal{Y}}$ denote a WFA over the $(+, \times)$ semiring accepting the set of all sequences of length $l$ with weight one and let $\mathcal{Y}_i$ denote the WFA accepting only $y_i$ with weight one. Then, by definition, the weighted transducer $\overline{\mathcal{Y}} \circ \mathcal{U} \circ \mathcal{Y}_i$ obtained by composition maps each sequence $y$ in $\Delta^l$ to $y_i$ with weight $\mathcal{U}(y, y_i)$. The WFA $\Pi_1(\overline{\mathcal{Y}} \circ \mathcal{U} \circ \mathcal{Y}_i)$ derived from that transducer by projection on the input (that is by removing output labels) is associating to each sequence $y$ weight $\mathcal{U}(y, y_i)$. We use weighted determinization [Mohri, 1997] to compute an equivalent deterministic WFA denote $\mathcal{M}$. As shown by Cortes et al. [2015][Theorem 3], $\mathcal{M}$ can be computed in polynomial time. $\mathcal{M}$ admits a unique path labeled with any sequence $y \in \Delta^l$ and the weight of that path is $\mathcal{U}(y, y_i)$. The weight of that accepting path is obtained by multiplying the weights of its transitions and that of the final state.

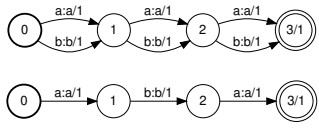

Figure 1: Illustration of the WFA $\overline{\mathcal{Y}}$ for $\Delta = \{a, b\}$ and $l = 3$, and the WFA $\mathcal{Y}_i$, where $y_i = aba$.

We now define a deterministic $p$-gram WFA $\mathcal{N}$ that accepts all sequences $y \in \Delta^l$ with each of its states $(\mathbf{z}', s)$ encoding a $(p-1)$-gram $\mathbf{z}'$ read to reach it and the position $s$ in the sequence $y$ at which it is reached. The transitions of $\mathcal{N}$ are therefore defined as follows with weight one:

$$\mathsf{E}_{\mathcal{N}} = \left\{ \left( \left( y_{s-p+1}^{s-1}, s-1 \right), a, 1, \left( y_{s-p+2}^{s-1} a, s \right) \right) : y \in \Delta^l, a \in \Delta, s \in [l] \right\}.$$

The initial state is $(\epsilon, 0)$ and the final states are those with the second element of the pair (the position) being $l$. Note that, by construction, $\mathcal{N}$ is deterministic. Then, the composition (or intersection) WFA $\mathcal{N} \circ \mathcal{M}$ still associates the same weight as $\mathcal{M}$ to each input string $y \in \Delta^l$. However, the states in that composition help us compute $\mathsf{L}(\mathbf{z}, s)$. In particular, for any $\mathbf{z} \in \Delta^p$ and $s \in [l]$, let $\mathsf{E}(\mathbf{z}, s)$ be the set of transitions of $\mathcal{N} \circ \mathcal{M}$ constructed by pairing the transition

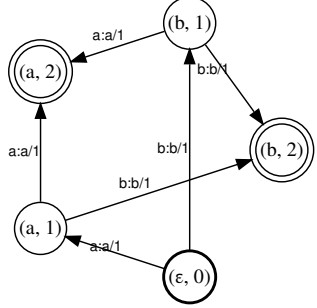

Figure 2: Illustration of the WFA $\mathcal{N}$ for $\Delta = \{a, b\}$, $p = 2$ and $l = 2$.

$\left( (\mathbf{z}_1^{p-1}, s-1), z_p, \omega(\mathbf{z}, s), (\mathbf{z}_2^p, s) \right)$ in $\mathcal{N}$ with a transition $(q_{\mathcal{M}}, z_p, \omega, q'_{\mathcal{M}})$ in $\mathcal{M}$. They admit the following form:

$$\mathsf{E}(\mathbf{z}, s) = \left\{ \left( (q_{\mathcal{N}}, q_{\mathcal{M}}), z_p, \omega, (q'_{\mathcal{N}}, q'_{\mathcal{M}}) \right) \in \mathsf{E}_{\mathcal{N} \circ \mathcal{M}} : q_{\mathcal{N}} = (\mathbf{z}_1^{p-1}, s-1) \right\}. \tag{14}$$

The WFA $\mathcal{N} \circ \mathcal{M}$ is deterministic as a composition of two deterministic WFAs. Thus, there is a unique path labeled with a sequence $y \in \Delta^l$ in $\mathcal{N} \circ \mathcal{M}$ and $y$ admits the substring $\mathbf{z}$ ending at position $s$ iff that path goes through a transition in $\mathsf{E}(\mathbf{z}, s)$ when reaching position $s$. Therefore, to compute $\mathsf{L}(\mathbf{z}, s)$, it suffices for us to compute the sum of the weights of all paths in $\mathcal{N} \circ \mathcal{M}$ going through a transition in $\mathsf{E}(\mathbf{z}, s)$. This can be done straightforwardly using the forward-backward algorithm or two single-source shortest-distance algorithm over the $(+, \times)$ semiring [Mohri, 2002a], one from the initial state, the other one from the final states. Since $\mathcal{N} \circ \mathcal{M}$ is acyclic and admits $O(l|\Delta|^p)$ transitions, we can compute all the quantities $\mathsf{L}(\mathbf{z}, s)$, $s \in [l]$ and $\mathbf{z} \in \Delta^p$, in time $O(l|\Delta|^p)$.

# 7 Conclusion

Our detailed study revealed shortcomings in commonly used surrogate loss functions and algorithms for structured prediction, prompting the introduction of new, strongly consistent alternatives. These findings not only enhance the theoretical and algorithmic foundations of structured prediction but also pave the way for the development of practical and effective solutions. In upcoming work, we will report an extensive empirical analysis of our algorithms. Our work provides tools and insights for future algorithm design in this domain, promising advancements in both theory and application.

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

# Contents of Appendix

# A    Related work

**Structured prediction and neural networks.** A variety of deep learning techniques have been used for structured prediction tasks, including a unified neural network architecture for natural language processing [Collobert et al., 2011], energy model-based structured prediction including structured prediction energy networks (SPENs) and various inference methods [Belanger and McCallum, 2016, Tu and Gimpel, 2018, Larsson et al., 2018, Tu and Gimpel, 2019, Tu et al., 2019, Graber and Schwing, 2019, Pan et al., 2020], attention networks incorporating richer structural distributions [Kim et al., 2017], tree-structured long short-term memory (LSTM) networks [Tai et al., 2015], sequence to sequence learning [Sutskever et al., 2014, Edunov et al., 2017], memory-reduced variant of best-first beam search [Meister et al., 2020], end-to-end learning for SPENs [Belanger et al., 2017], conditional generative flow [Lu and Huang, 2020], proximal methods [Wang et al., 2016], CRF using deep features [Jaderberg et al., 2014, Huang et al., 2015, Chen et al., 2014, Schwing and Urtasun, 2015, Chen et al., 2015], non-iterative feed-forward predictors [Stoyanov et al., 2011, Domke, 2013, Kunisch and Pock, 2013, Hershey et al., 2014, Li and Zemel, 2014, Belharbi et al., 2018, Zheng et al., 2015], deep value network [Gygli et al., 2017], fully convolutional networks [Long et al., 2015], constraint reasoning tool [Dragone et al., 2021, Jiang et al., 2022], multi-label box model[Patel et al., 2022], probabilistic deep networks[Zheng and Pronobis, 2019, Jang et al., 2023], autoregressive methods [Liu et al., 2022], nonlinear output transformations and embedding [Graber et al., 2018, Brogat-Motte et al., 2020], smoothing methods [Pillutla et al., 2018] and structural training [Choi et al., 2016, Ahmad et al., 2023]. Let us also mention ensemble algorithms for structured prediction algorithms [Cortes, Kuznetsov, and Mohri, 2014a,b], which can be used to combine several algorithms for this problem.

**Consistency in structured prediction.** Here, we discuss in detail previous work on consistency in structured prediction [Osokin et al., 2017, Ciliberto et al., 2016, Blondel, 2019, Nowak et al., 2020, 2022, Ciliberto et al., 2019, 2020, Nowak-Vila et al., 2019, Nowak et al., 2019, Cabannes et al., 2021, Cabannes et al., 2020, Corro, 2023].

Osokin et al. [2017], Nowak et al. [2020] and Nowak et al. [2022] pointed out that the max-margin Markov networks (M3N), or more generally structural SVMs may not be Bayes-consistent. Instead, Osokin et al. [2017] proposed the first Bayes-consistent surrogate loss in the structured prediction setting, the quadratic surrogate (QS) loss. A general theory of QS was further developed in [Nowak-Vila et al., 2019, Nowak et al., 2019]. However, as pointed out in Section 1, the quadratic surrogate loss formulation casts the structured prediction problem as a regression problem and is not a typical formulation even in the binary classification case.

Nowak et al. [2020] proposed a consistent method called *max-min margin Markov networks (M4N)* derived from first principles for binary SVM. However, this method is restricted to SVM-type loss functions. Instead, we propose broad families of surrogate losses, which can be naturally derived from common multi-class losses including the logistic loss.

Nowak et al. [2022] addressed the inconsistency of Max-Margin loss in structured prediction by introducing the notion of Restricted-Max-Margin, where the maximization is performed over a subset of the original domain. Their method is based on an implicit embedding [Finocchiaro et al., 2019]; a general framework for structured prediction has been further developed by Ciliberto et al. [2020]. However, these methods are only applied to polyhedral-type surrogates which are not as smooth as the logistic loss. Thus, the resulting surrogate losses may not be favorable from the optimization point of view. Instead, our novel families of surrogate losses are very general and can be smooth, including a new structured logistic loss, for which we describe efficient gradient computation algorithms.

Ciliberto et al. [2016] focused on a least squares surrogate loss function and corresponding framework. In this framework, the structured prediction problem is cast as a regression problem. They derived a regularization approach to structured prediction from the least squares surrogate loss and proved the Bayes-consistency of that approach. Ciliberto et al. [2019] focused on a local structure-adapted framework for structured prediction. They proposed a novel structured prediction algorithm that adaptively leverages locality in the learning problem. Ciliberto et al. [2020] developed a general framework for structured prediction based on implicit embedding. Their methods lead to polyhedral-type surrogates losses that benefit from Bayes-consistency.

On the other hand, our work presents an extensive study of surrogate losses for structured prediction supported by $\mathcal{H}$-consistency bounds. Different from the surrogate loss studied in the previous work,

the formulations of our proposed surrogate losses including structured comp-sum losses and structured constrained losses are completely novel and do not cast structured prediction problems as a regression problem. Furthermore, we prove stronger consistency guarantees that imply Bayes-consistency for these new proposed families of surrogate loss.

Other related work on structured prediction includes: projection-based losses for structured prediction [Blondel, 2019]; fast convergence rates for general structured prediction problems [Cabannes et al., 2021]; a unified framework for dealing with partial labelling [Cabannes et al., 2020]; and an analysis of the inconsistency of separable negative log-likelihood losses for structured prediction [Corro, 2023].

# B  Proof of Lemma 3

**Lemma 3.** *The best-in-class conditional error and the conditional regret for a target loss* $\mathsf{L}$ *in structured prediction can be expressed as follows:*

$$\mathcal{C}_{\mathsf{L},\mathcal{H}}^{*}(x) = \min_{y' \in \mathsf{H}(x)} \sum_{y \in \mathcal{Y}} p(x,y)\ell(y',y)$$

$$\Delta\mathcal{C}_{\mathsf{L},\mathcal{H}}(h,x) = \sum_{y \in \mathcal{Y}} p(x,y)\ell(\mathsf{h}(x),y) - \min_{y' \in \mathsf{H}(x)} \sum_{y \in \mathcal{Y}} p(x,y)\ell(y',y).$$

*Proof.* By the definition, the conditional L-risk can be expressed as follows:

$$\mathcal{C}_{\mathsf{L}}(h,x) = \sum_{y \in \mathcal{Y}} p(x,y)\ell(\mathsf{h}(x),y). \tag{15}$$

Since $\{\mathsf{h}(x) : h \in \mathcal{H}\} = \mathsf{H}(x)$, the best-in-class conditional error can be expressed as follows:

$$\mathcal{C}_{\mathsf{L},\mathcal{H}}^{*}(x) = \min_{y' \in \mathsf{H}(x)} \sum_{y \in \mathcal{Y}} p(x,y)\ell(y',y),$$

which proves the first part of the lemma. By the definition,

$$\Delta\mathcal{C}_{\mathsf{L},\mathcal{H}}(h,x) = \mathcal{C}_{\mathsf{L}}(h,x) - \mathcal{C}_{\mathsf{L},\mathcal{H}}^{*}(x) = \sum_{y \in \mathcal{Y}} p(x,y)\ell(\mathsf{h}(x),y) - \min_{y' \in \mathsf{H}(x)} \sum_{y \in \mathcal{Y}} p(x,y)\ell(y',y).$$

$\square$

# C  Proofs for structured max losses

**Theorem 4** (**Negative results of** $\mathsf{L}^{\max}$). *Assume that $n > 2$ and that $\Phi_u(v)$ is convex and non-increasing for $u = 1$. Then, the max structured loss $\mathsf{L}^{\max}$ is not Bayes-consistent.*

*Proof.* For the structured max loss $\mathsf{L}^{\max}$, the conditional $\mathsf{L}^{\max}$-risk can be expressed as follows:

$$\mathcal{C}_{\mathsf{L}^{\max}}(h,x) = \sum_{y \in \mathcal{Y}} p(x,y) \max_{y' \neq y} \Phi_{\ell(y',y)}(h(x,y) - h(x,y')).$$

Take $\ell(y',y) = \mathbb{1}_{y \neq y'}$ to be the zero-one loss. Since $\ell(y',y) = 1$ for any $y \neq y'$, the conditional $\mathsf{L}^{\max}$-risk can be reformulated as follows:

$$\mathcal{C}_{\mathsf{L}^{\max}}(h,x) = \sum_{y \in \mathcal{Y}} p(x,y) \max_{y' \neq y} \Phi_1(h(x,y) - h(x,y')).$$

Consider the distribution that supports on a singleton domain $\{x\}$. Take $y_1 \neq y_2 \in \mathcal{Y}$ such that $y_1 \neq n$ and $y_2 \neq n$. We define the conditional distribution as $p(x,y_1) = p(x,y_2) = \frac{1}{2}$ and $p(x,y) = 0$ for other $y \in \mathcal{Y}$. Then, by using the fact that $\Phi_1(v)$ is convex and non-increasing, we obtain

$$\begin{aligned}
\mathcal{R}_{\mathsf{L}^{\max}}(h) = \mathcal{C}_{\mathsf{L}^{\max}}(h,x) &= \frac{1}{2} \max_{y' \neq y_1} \Phi_1(h(x,y_1) - h(x,y')) + \frac{1}{2} \max_{y' \neq y_2} \Phi_1(h(x,y_2) - h(x,y')) \\
&= \frac{1}{2} \Phi_1\Big(h(x,y_1) - \max_{y' \neq y_1} h(x,y')\Big) + \frac{1}{2} \Phi_1\Big(h(x,y_2) - \max_{y' \neq y_2} h(x,y')\Big) \\
&\hspace{6cm} (\Phi_1(v) \text{ is non-increasing}) \\
&\geq \Phi_1\Big(\frac{1}{2}h(x,y_1) - \frac{1}{2} \max_{y' \neq y_2} h(x,y') + \frac{1}{2}h(x,y_2) - \frac{1}{2} \max_{y' \neq y_1} h(x,y')\Big) \\
&\hspace{6cm} (\Phi_1(v) \text{ is convex}) \\
&\geq \Phi_1(0) \hspace{3.5cm} (\Phi_1(v) \text{ is non-increasing})
\end{aligned}$$

where the equality can be achieved by $h^* \in \mathcal{H}$, defined as $h^*(x,1) = h^*(x,2) = \ldots = h^*(x,n)$. Therefore, $h^*$ is a Bayes classifier of the structured max loss. Note that $\mathsf{h}^*(x) = n$. However, by Lemma 3, in such a case, the Bayes classifier $\mathsf{h}_\ell^*$ of the target loss satisfies that

$$\mathsf{h}_\ell^*(x) = \operatorname*{argmin}_{y' \in \mathcal{Y}} \sum_{y \in \mathcal{Y}} p(x,y)\ell(y',y) = \operatorname*{argmin}_{y' \in \mathcal{Y}}(p(x,y_1)\mathbb{1}_{y' \neq y_1} + p(x,y_2)\mathbb{1}_{y' \neq y_2}) = y_1 \text{ or } y_2.$$

Thus, we obtain $\mathsf{h}^* \neq \mathsf{h}_\ell^*$. Therefore, $\mathsf{L}^{\max}$ is not Bayes-consistent. $\qquad\square$

## D   Proofs for Voted Conditional Random Field

**Theorem 5** (**Negative result of $\mathsf{L}^{\mathbf{VCRF}}$**). *The Voted Conditional Random Field $\mathsf{L}^{\mathrm{VCRF}}$ is not Bayes-consistent.*

*Proof.* For the Voted Conditional Random Field $\mathsf{L}^{\mathrm{VCRF}}$, the conditional $\mathsf{L}^{\mathrm{VCRF}}$-risk can be expressed as follows:

$$\mathcal{C}_{\mathsf{L}^{\mathrm{VCRF}}}(h,x) = \sum_{y \in \mathcal{Y}} p(x,y)\log\left(\sum_{y' \in \mathcal{Y}} e^{\ell(y,y')+h(x,y')-h(x,y)}\right).$$

Consider the distribution that supports on a singleton domain $\{x\}$. Note that $\mathcal{R}_{\mathsf{L}^{\mathrm{VCRF}}} = \mathcal{C}_{\mathsf{L}^{\mathrm{VCRF}}}$ is convex with respect to $h(x,y), y = 1,\ldots,n$. To find the global minimum, we will differentiate $\mathcal{C}_{\mathsf{L}^{\mathrm{VCRF}}}$ with respect to $h(x,y)$ for any $y \in \mathcal{Y}$ and setting the derivatives to zero. Thus, we obtain for any $y \in \mathcal{Y}$,

$$p(x,y)\frac{-\sum_{y' \neq y} e^{\ell(y,y')+h(x,y')-h(x,y)}}{\sum_{y' \in \mathcal{Y}} e^{\ell(y,y')+h(x,y')-h(x,y)}} + \sum_{y' \neq y} p(x,y')\frac{e^{\ell(y',y)+h(x,y)-h(x,y')}}{\sum_{y'' \in \mathcal{Y}} e^{\ell(y',y'')+h(x,y'')-h(x,y')}} = 0. \quad (16)$$

Using the fact that $\sum_{y' \neq y} e^{\ell(y,y')+h(x,y')-h(x,y)} = \sum_{y' \in \mathcal{Y}} e^{\ell(y,y')+h(x,y')-h(x,y)} - e^{\ell(y,y)+h(x,y)-h(x,y)}$ to further simplify the LHS of (16), we obtain for any $y \in \mathcal{Y}$,

$$p(x,y) = \sum_{y' \in \mathcal{Y}} p(x,y')\frac{e^{\ell(y',y)+h(x,y)-h(x,y')}}{\sum_{y'' \in \mathcal{Y}} e^{\ell(y',y'')+h(x,y'')-h(x,y')}} = \sum_{y' \in \mathcal{Y}} p(x,y')\frac{e^{\ell(y',y)+h(x,y)}}{\sum_{y'' \in \mathcal{Y}} e^{\ell(y',y'')+h(x,y'')}}. \quad (17)$$

Consider a target loss function $\mathsf{L}$ such that $e^{\ell(y,y')} = \Phi_y \Phi_{y'}$, that is $\ell(y,y') = \log(\Phi_y) + \log(\Phi_{y'})$, where $\Phi_y$ is a function mapping from $\mathcal{Y}$ to $\mathbb{R}_+$. For this special choice of the target loss function, the expression of $\ell(y,y')$ decouple and (17) can be simplified to

$$p(x,y) = \sum_{y' \in \mathcal{Y}} p(x,y')\frac{\Phi_y \Phi_{y'} e^{h(x,y)}}{\sum_{y'' \in \mathcal{Y}} \Phi_{y'}\Phi_{y''} e^{h(x,y'')}} = \frac{\Phi_y e^{h(x,y)}}{\sum_{y' \in \mathcal{Y}} \Phi_{y'} e^{h(x,y')}}. \quad (18)$$

Therefore, for the Bayes classifier $h^*$ of Voted Conditional Random Field, by (18), we have

$$\frac{\Phi_y e^{h^*(x,1)}}{p(x,1)} = \ldots = \frac{\Phi_y e^{h^*(x,n)}}{p(x,n)}$$

which implies that

$$\mathsf{h}^*(x) = \operatorname*{argmax}_{y' \in \mathcal{Y}} h^*(x,y') = \operatorname*{argmax}_{y' \in \mathcal{Y}} e^{h^*(x,y')} = \operatorname*{argmax}_{y' \in \mathcal{Y}} \frac{p(x,y')}{\Phi_{y'}}.$$

However, by Lemma 3, in such a case, the Bayes classifier $\mathsf{h}_\ell^*$ of the target loss satisfies that

$$\mathsf{h}_\ell^*(x) = \operatorname*{argmin}_{y' \in \mathcal{Y}} \sum_{y \in \mathcal{Y}} p(x,y)\ell(y',y) = \operatorname*{argmin}_{y' \in \mathcal{Y}} \sum_{y \in \mathcal{Y}} p(x,y)(\log(\Phi_y) + \log(\Phi_{y'})) = \operatorname*{argmin}_{y' \in \mathcal{Y}} \Phi_{y'}.$$

Thus, we obtain $\mathsf{h}^* \neq \mathsf{h}_\ell^*$ in general. Indeed, consider the case where $p(x,y) = \frac{\Phi_y^2}{\sum_{k=1}^n \Phi_k^2}, y \in \mathcal{Y}$. Then, $\mathsf{h}^*(x) = \operatorname{argmax}_{y' \in \mathcal{Y}} \frac{\Phi_{y'}}{\sum_{k=1}^n \Phi_k^2} = \operatorname{argmax}_{y' \in \mathcal{Y}} \Phi_{y'} \neq \operatorname{argmin}_{y' \in \mathcal{Y}} \Phi_{y'} = \mathsf{h}_\ell^*(x)$ when $\{\Phi_y : y \in \mathcal{Y}\}$ are not equal. Therefore, $\mathsf{L}^{\mathrm{VCRF}}$ is not Bayes-consistent. $\qquad\square$

# E    Proofs for structured comp-sum losses

## E.1    Structured logistic loss

**Theorem 11** ($\mathcal{H}$**-consistency bound of** $\mathsf{L}_{\log}^{\mathrm{comp}}$). *Assume that $\mathcal{H}$ is symmetric and complete. Then, for any target loss $\ell$, hypothesis $h \in \mathcal{H}$ and any distribution,*

$$\mathcal{R}_{\mathsf{L}}(h) - \mathcal{R}_{\mathsf{L},\mathcal{H}}^* \leq 2\Big(\mathcal{R}_{\mathsf{L}_{\log}^{\mathrm{comp}}}(h) - \mathcal{R}_{\mathsf{L}_{\log}^{\mathrm{comp}},\mathcal{H}}^* + \mathcal{M}_{\mathsf{L}_{\log}^{\mathrm{comp}},\mathcal{H}}\Big)^{\frac{1}{2}} - \mathcal{M}_{\mathsf{L},\mathcal{H}}. \tag{19}$$

*Proof.* For the comp-sum structured loss $\mathsf{L}_{\log}^{\mathrm{comp}}$, the conditional $\mathsf{L}_{\log}^{\mathrm{comp}}$-risk can be expressed as follows:

$$\mathcal{C}_{\mathsf{L}_{\log}^{\mathrm{comp}}}(h, x) = -\sum_{y \in \mathcal{Y}} p(x,y) \sum_{y' \in \mathcal{Y}} \overline{\ell}(y', y) \log\left(\frac{e^{h(x,y')}}{\sum_{y'' \in \mathcal{Y}} e^{h(x,y'')}}\right)$$

$$= -\sum_{y' \in \mathcal{Y}} \log\left(\frac{e^{h(x,y')}}{\sum_{y'' \in \mathcal{Y}} e^{h(x,y'')}}\right) \sum_{y \in \mathcal{Y}} p(x,y)\overline{\ell}(y', y)$$

$$= -\sum_{y' \in \mathcal{Y}} \log(\mathcal{S}(x,y'))\overline{q}(x,y'),$$

where we denote by $\overline{q}(x,y') = \sum_{y \in \mathcal{Y}} p(x,y)\overline{\ell}(y',y) \in [0,1]$ and $\mathcal{S}(x,y) = \frac{e^{h(x,y)}}{\sum_{y'' \in \mathcal{Y}} e^{h(x,y'')}} \in [0,1]$ with the constraint that $\sum_{y \in \mathcal{Y}} \mathcal{S}(x,y) = 1$. Let $y_{\min} = \operatorname{argmin}_{y' \in \mathcal{Y}} \sum_{y \in \mathcal{Y}} p(x,y)\ell(y',y)$, where we choose the label with the highest index under the natural ordering of labels as the tie-breaking strategy. For any $h \in \mathcal{H}$ such that $\mathsf{h}(x) \neq y_{\min}$ and $x \in \mathcal{X}$, by the symmetry and completeness of $\mathcal{H}$, we can always find a family of hypotheses $\{\overline{h}_\mu : \mu \in [-\mathcal{S}(x,y_{\min}), \mathcal{S}(x,\mathsf{h}(x))]\} \subset \mathcal{H}$ such that $\overline{\mathcal{S}}_\mu(x,\cdot) = \frac{e^{h_\mu(x,\cdot)}}{\sum_{y' \in \mathcal{Y}} e^{h_\mu(x,y')}}$ take the following values:

$$\overline{\mathcal{S}}_\mu(x,y) = \begin{cases} \mathcal{S}(x,y) & \text{if } y \notin \{y_{\min}, \mathsf{h}(x)\} \\ \mathcal{S}(x,y_{\min}) + \mu & \text{if } y = \mathsf{h}(x) \\ \mathcal{S}(x,\mathsf{h}(x)) - \mu & \text{if } y = y_{\min}. \end{cases} \tag{20}$$

Note that $\overline{\mathcal{S}}_\mu$ satisfies the constraint:

$$\sum_{y \in \mathcal{Y}} \overline{\mathcal{S}}_\mu(x,y) = \sum_{y \in \mathcal{Y}} \mathcal{S}(x,y) = 1, \ \forall \mu \in [-\mathcal{S}(x,y_{\min}), \mathcal{S}(x,\mathsf{h}(x))].$$

By (20) and using the fact that $\mathsf{H}(x) = \mathcal{Y}$ when $\mathcal{H}$ is symmetric, we obtain

$$\Delta\mathcal{C}_{\mathsf{L}^{\mathrm{comp}},\mathcal{H}}(h,x) = \mathcal{C}_{\mathsf{L}^{\mathrm{comp}}}(h,x) - \mathcal{C}_{\mathsf{L}^{\mathrm{comp}}}^*(\mathcal{H},x)$$

$$\geq \mathcal{C}_{\mathsf{L}^{\mathrm{comp}}}(h,x) - \inf_{\mu \in \mathbb{R}} \mathcal{C}_{\mathsf{L}^{\mathrm{comp}}}(\overline{h}_\mu, x)$$

$$= \sup_{\mu \in [-\mathcal{S}(x,y_{\min}), \mathcal{S}(x,\mathsf{h}(x))]} \Big\{ \overline{q}(x,y_{\min})[-\log(\mathcal{S}(x,y_{\min})) + \log(\mathcal{S}(x,\mathsf{h}(x)) - \mu)]$$

$$+ \overline{q}(x,\mathsf{h}(x))[-\log(\mathcal{S}(x,\mathsf{h}(x))) + \log(\mathcal{S}(x,y_{\min}) + \mu)] \Big\}.$$

Differentiating with respect to $\mu$ yields the optimal value $\mu^* = \frac{\overline{q}(x,\mathsf{h}(x))\mathcal{S}(x,\mathsf{h}(x)) - \overline{q}(x,y_{\min})\mathcal{S}(x,y_{\min})}{\overline{q}(x,y_{\min}) + \overline{q}(x,\mathsf{h}(x))}$. Plugging in that value gives:

$$\Delta\mathcal{C}_{\mathsf{L}^{\mathrm{comp}},\mathcal{H}}(h,x) \geq \overline{q}(x,y_{\min}) \log \frac{(\mathcal{S}(x,\mathsf{h}(x)) + \mathcal{S}(x,y_{\min}))\overline{q}(x,y_{\min})}{\mathcal{S}(x,y_{\min})(\overline{q}(x,y_{\min}) + \overline{q}(x,\mathsf{h}(x)))}$$

$$+ \overline{q}(x,\mathsf{h}(x)) \log \frac{(\mathcal{S}(x,\mathsf{h}(x)) + \mathcal{S}(x,y_{\min}))\overline{q}(x,\mathsf{h}(x))}{\mathcal{S}(x,\mathsf{h}(x))(\overline{q}(x,y_{\min}) + \overline{q}(x,\mathsf{h}(x)))}.$$

Differentiating with respect to $\mathcal{S}$ shows that the minimum is attained for $\mathcal{S}(x, \mathsf{h}(x)) = \mathcal{S}(x, y_{\min})$, which gives:

$$\Delta\mathcal{C}_{\mathsf{L}^{\mathrm{comp}},\mathcal{H}}(h, x) \geq \overline{q}(x, y_{\min}) \log \frac{2\overline{q}(x, y_{\min})}{\overline{q}(x, y_{\min}) + \overline{q}(x, \mathsf{h}(x))} + \overline{q}(x, \mathsf{h}(x)) \log \frac{2\overline{q}(x, \mathsf{h}(x))}{\overline{q}(x, y_{\min}) + \overline{q}(x, \mathsf{h}(x))}$$

$$\geq \frac{\left(\overline{q}(x, \mathsf{h}(x)) - \overline{q}(x, y_{\min})\right)^2}{2(\overline{q}(x, \mathsf{h}(x)) + \overline{q}(x, y_{\min}))}$$

$$(a \log \tfrac{2a}{a+b} + b \log \tfrac{2b}{a+b} \geq \tfrac{(a-b)^2}{2(a+b)}, \forall a, b \in [0, 1] \text{ [Mohri et al., 2018, Proposition E.7]})$$

$$\geq \frac{\left(\overline{q}(x, \mathsf{h}(x)) - \overline{q}(x, y_{\min})\right)^2}{4} \qquad\qquad (0 \leq \overline{q}(x, \mathsf{h}(x)) + \overline{q}(x, y_{\min}) \leq 2)$$

$$= \frac{\left(\sum_{y \in \mathcal{Y}} p(x, y)\ell(\mathsf{h}(x), y) - \sum_{y \in \mathcal{Y}} p(x, y)\ell(y_{\min}, y)\right)^2}{4}$$

$$= \frac{1}{4}\Delta\mathcal{C}_{\mathsf{L},\mathcal{H}}(h, x)^2. \qquad\qquad (\text{by Lemma 3 and } \mathsf{H}(x) = \mathcal{Y})$$

Since the function $t \mapsto \frac{t^2}{4}$ is convex, by Jensen's inequality, we obtain for any hypothesis $h \in \mathcal{H}$ and any distribution,

$$\frac{\left(\mathcal{R}_{\mathsf{L}}(h) - \mathcal{R}^*_{\mathsf{L},\mathcal{H}} + \mathcal{M}_{\mathsf{L},\mathcal{H}}\right)^2}{4} = \frac{\left(\mathbb{E}_X[\Delta\mathcal{C}_{\mathsf{L},\mathcal{H}}(h, x)]\right)^2}{4}$$

$$\leq \mathbb{E}_X\left[\frac{\Delta\mathcal{C}_{\mathsf{L},\mathcal{H}}(h, x)^2}{4}\right]$$

$$\leq \mathbb{E}_X[\Delta\mathcal{C}_{\mathsf{L}^{\mathrm{comp}},\mathcal{H}}(h, x)]$$

$$= \mathcal{R}_{\mathsf{L}^{\mathrm{comp}}}(h) - \mathcal{R}^*_{\mathsf{L}^{\mathrm{comp}},\mathcal{H}} + \mathcal{M}_{\mathsf{L}^{\mathrm{comp}},\mathcal{H}},$$

which leads to

$$\mathcal{R}_{\mathsf{L}}(h) - \mathcal{R}^*_{\mathsf{L},\mathcal{H}} \leq 2\left(\mathcal{R}_{\mathsf{L}^{\mathrm{comp}}_{\log}}(h) - \mathcal{R}^*_{\mathsf{L}^{\mathrm{comp}}_{\log},\mathcal{H}} + \mathcal{M}_{\mathsf{L}^{\mathrm{comp}}_{\log},\mathcal{H}}\right)^{\frac{1}{2}} - \mathcal{M}_{\mathsf{L},\mathcal{H}}.$$

$\square$

### E.2 Structured sum-exponential loss

**Theorem 12** ($\mathcal{H}$-**consistency bound of** $\mathsf{L}^{\mathrm{comp}}_{\exp}$). *Assume that $\mathcal{H}$ is symmetric and complete. Then, for any target loss $\ell$, hypothesis $h \in \mathcal{H}$ and any distribution,*

$$\mathcal{R}_{\mathsf{L}}(h) - \mathcal{R}^*_{\mathsf{L},\mathcal{H}} \leq 2\left(\mathcal{R}_{\mathsf{L}^{\mathrm{comp}}_{\exp}}(h) - \mathcal{R}^*_{\mathsf{L}^{\mathrm{comp}}_{\exp},\mathcal{H}} + \mathcal{M}_{\mathsf{L}^{\mathrm{comp}}_{\exp},\mathcal{H}}\right)^{\frac{1}{2}} - \mathcal{M}_{\mathsf{L},\mathcal{H}}. \qquad (21)$$

*Proof.* For the comp-sum structured loss $\mathsf{L}^{\mathrm{comp}}_{\exp}$, the conditional $\mathsf{L}^{\mathrm{comp}}_{\exp}$-risk can be expressed as follows:

$$\mathcal{C}_{\mathsf{L}^{\mathrm{comp}}_{\exp}}(h, x) = \sum_{y \in \mathcal{Y}} p(x, y) \sum_{y' \in \mathcal{Y}} \overline{\ell}(y', y) \sum_{y'' \neq y'} e^{h(x, y'') - h(x, y')} = \sum_{y' \in \mathcal{Y}} \left(\frac{1}{\mathcal{S}(x, y')} - 1\right)\overline{q}(x, y'),$$

where we denote by $\overline{q}(x, y') = \sum_{y \in \mathcal{Y}} p(x, y)\overline{\ell}(y', y) \in [0, 1]$ and $\mathcal{S}(x, y) = \frac{e^{h(x, y)}}{\sum_{y'' \in \mathcal{Y}} e^{h(x, y'')}} \in [0, 1]$ with the constraint that $\sum_{y \in \mathcal{Y}} \mathcal{S}(x, y) = 1$. Let $y_{\min} = \operatorname{argmin}_{y' \in \mathcal{Y}} \sum_{y \in \mathcal{Y}} p(x, y)\ell(y', y)$, where we choose the label with the highest index under the natural ordering of labels as the tie-breaking strategy. For any $h \in \mathcal{H}$ such that $\mathsf{h}(x) \neq y_{\min}$ and $x \in \mathcal{X}$, by the symmetry and completeness of $\mathcal{H}$, we can always find a family of hypotheses $\left\{\overline{h}_\mu : \mu \in \left[-\mathcal{S}(x, y_{\min}), \mathcal{S}(x, \mathsf{h}(x))\right]\right\} \subset \mathcal{H}$ such that $\overline{\mathcal{S}}_\mu(x, \cdot) = \frac{e^{h_\mu(x, \cdot)}}{\sum_{y' \in \mathcal{Y}} e^{h_\mu(x, y')}}$ take the following values:

$$\overline{\mathcal{S}}_\mu(x, y) = \begin{cases} \mathcal{S}(x, y) & \text{if } y \notin \{y_{\min}, \mathsf{h}(x)\} \\ \mathcal{S}(x, y_{\min}) + \mu & \text{if } y = \mathsf{h}(x) \\ \mathcal{S}(x, \mathsf{h}(x)) - \mu & \text{if } y = y_{\min}. \end{cases} \qquad (22)$$

Note that $\overline{\mathcal{S}}_\mu$ satisfies the constraint:

$$\sum_{y\in\mathcal{Y}}\overline{\mathcal{S}}_\mu(x,y) = \sum_{y\in\mathcal{Y}}\mathcal{S}(x,y) = 1,\ \forall\mu\in\left[-\mathcal{S}(x,y_{\min}),\mathcal{S}(x,\mathsf{h}(x))\right].$$

By (22) and using the fact that $\mathsf{H}(x) = \mathcal{Y}$ when $\mathcal{H}$ is symmetric, we obtain

$$\Delta\mathcal{C}_{\mathsf{L}^{\mathrm{comp}},\mathcal{H}}(h,x)$$
$$= \mathcal{C}_{\mathsf{L}^{\mathrm{comp}}}(h,x) - \mathcal{C}^*_{\mathsf{L}^{\mathrm{comp}}}(\mathcal{H},x)$$
$$\geq \mathcal{C}_{\mathsf{L}^{\mathrm{comp}}}(h,x) - \inf_{\mu\in\mathbb{R}}\mathcal{C}_{\mathsf{L}^{\mathrm{comp}}}(\overline{h}_\mu,x)$$
$$= \sup_{\mu\in[-\mathcal{S}(x,y_{\min}),\mathcal{S}(x,\mathsf{h}(x))]}\left\{\overline{q}(x,y_{\min})\left[\frac{1}{\mathcal{S}(x,y_{\min})} - \frac{1}{\mathcal{S}(x,\mathsf{h}(x))-\mu}\right]\right.$$
$$\left.+ \overline{q}(x,\mathsf{h}(x))\left[\frac{1}{\mathcal{S}(x,\mathsf{h}(x))} - \frac{1}{\mathcal{S}(x,y_{\min})+\mu}\right]\right\}$$
$$= \frac{\overline{q}(x,y_{\min})}{\mathcal{S}(x,y_{\min})} + \frac{\overline{q}(x,\mathsf{h}(x))}{\mathcal{S}(x,\mathsf{h}(x))} - \frac{\left(\sqrt{\overline{q}(x,y_{\min})}+\sqrt{\overline{q}(x,\mathsf{h}(x))}\right)^2}{\mathcal{S}(x,y_{\min})+\mathcal{S}(x,\mathsf{h}(x))}$$

(differentiating with respect to $\mu$ to optimize, optimal $\mu^* = \frac{\sqrt{\overline{q}(x,\mathsf{h}(x))}\mathcal{S}(x,\mathsf{h}(x))-\sqrt{\overline{q}(x,y_{\min})}\mathcal{S}(x,y_{\min})}{\sqrt{\overline{q}(x,y_{\min})}+\sqrt{\overline{q}(x,\mathsf{h}(x))}}$)

$$\geq \left(\sqrt{\overline{q}(x,y_{\min})}-\sqrt{\overline{q}(x,\mathsf{h}(x))}\right)^2$$

(differentiating with respect to $\mathcal{S}$ to minimize, minimum is attained when $\mathcal{S}(x,\mathsf{h}(x)) = \mathcal{S}(x,y_{\min}) = \frac{1}{2}$)

$$\geq \frac{(\overline{q}(x,\mathsf{h}(x)) - \overline{q}(x,y_{\min}))^2}{\left(\sqrt{\overline{q}(x,\mathsf{h}(x))}+\sqrt{\overline{q}(x,y_{\min})}\right)^2}$$

$$\geq \frac{(\overline{q}(x,\mathsf{h}(x)) - \overline{q}(x,y_{\min}))^2}{4} \qquad (\sqrt{a}+\sqrt{b}\leq 2,\forall a,b\in[0,1],a+b\leq 2)$$

$$= \frac{\left(\sum_{y\in\mathcal{Y}}p(x,y)\ell(\mathsf{h}(x),y) - \sum_{y\in\mathcal{Y}}p(x,y)\ell(y_{\min},y)\right)^2}{4}$$

$$= \frac{1}{4}\Delta\mathcal{C}_{\mathsf{L},\mathcal{H}}(h,x)^2. \qquad\qquad (\text{by Lemma } 3 \text{ and } \mathsf{H}(x) = \mathcal{Y})$$

Since the function $t\mapsto\frac{t^2}{4}$ is convex, by Jensen's inequality, we obtain for any hypothesis $h\in\mathcal{H}$ and any distribution,

$$\frac{\left(\mathcal{R}_{\mathsf{L}}(h) - \mathcal{R}^*_{\mathsf{L},\mathcal{H}} + \mathcal{M}_{\mathsf{L},\mathcal{H}}\right)^2}{4} = \frac{(\mathbb{E}_X[\Delta\mathcal{C}_{\mathsf{L},\mathcal{H}}(h,x)])^2}{4}$$
$$\leq \mathbb{E}_X\left[\frac{\Delta\mathcal{C}_{\mathsf{L},\mathcal{H}}(h,x)^2}{4}\right]$$
$$\leq \mathbb{E}_X\left[\Delta\mathcal{C}_{\mathsf{L}^{\mathrm{comp}},\mathcal{H}}(h,x)\right]$$
$$= \mathcal{R}_{\mathsf{L}^{\mathrm{comp}}}(h) - \mathcal{R}^*_{\mathsf{L}^{\mathrm{comp}},\mathcal{H}} + \mathcal{M}_{\mathsf{L}^{\mathrm{comp}},\mathcal{H}},$$

which leads to

$$\mathcal{R}_{\mathsf{L}}(h) - \mathcal{R}^*_{\mathsf{L},\mathcal{H}} \leq 2\left(\mathcal{R}_{\mathsf{L}^{\mathrm{comp}}_{\mathrm{exp}}}(h) - \mathcal{R}^*_{\mathsf{L}^{\mathrm{comp}}_{\mathrm{exp}},\mathcal{H}} + \mathcal{M}_{\mathsf{L}^{\mathrm{comp}}_{\mathrm{exp}},\mathcal{H}}\right)^{\frac{1}{2}} - \mathcal{M}_{\mathsf{L},\mathcal{H}}.$$

$\square$

### E.3 Structured generalized cross-entropy loss

**Theorem 13** ($\mathcal{H}$-**consistency bound of** $\mathsf{L}^{\mathrm{comp}}_{\mathrm{gce}}$). *Assume that $\mathcal{H}$ is symmetric and complete. Then, for any target loss $\ell$, hypothesis $h\in\mathcal{H}$ and any distribution,*

$$\mathcal{R}_{\mathsf{L}}(h) - \mathcal{R}^*_{\mathsf{L},\mathcal{H}} \leq 2n^{\frac{\alpha}{2}}\left(\mathcal{R}_{\mathsf{L}^{\mathrm{comp}}_{\mathrm{gce}}}(h) - \mathcal{R}^*_{\mathsf{L}^{\mathrm{comp}}_{\mathrm{gce}},\mathcal{H}} + \mathcal{M}_{\mathsf{L}^{\mathrm{comp}}_{\mathrm{gce}},\mathcal{H}}\right)^{\frac{1}{2}} - \mathcal{M}_{\mathsf{L},\mathcal{H}}. \qquad (23)$$

*Proof.* For the comp-sum structured loss $\mathsf{L}^{\mathrm{comp}}_{\mathrm{gce}}$, the conditional $\mathsf{L}^{\mathrm{comp}}_{\mathrm{gce}}$-risk can be expressed as follows:

$$
\begin{aligned}
\mathfrak{C}_{\mathsf{L}^{\mathrm{comp}}_{\mathrm{gce}}}(h, x) &= \sum_{y \in \mathcal{Y}} p(x, y) \sum_{y' \in \mathcal{Y}} \overline{\ell}(y', y) \frac{1}{\alpha} \left( 1 - \left( \frac{e^{h(x, y')}}{\sum_{y'' \in \mathcal{Y}} e^{h(x, y'')}} \right)^{\alpha} \right) \\
&= \frac{1}{\alpha} \sum_{y' \in \mathcal{Y}} \left( 1 - \left( \frac{e^{h(x, y')}}{\sum_{y'' \in \mathcal{Y}} e^{h(x, y'')}} \right)^{\alpha} \right) \sum_{y \in \mathcal{Y}} p(x, y) \overline{\ell}(y', y) \\
&= \frac{1}{\alpha} \sum_{y' \in \mathcal{Y}} (1 - \mathcal{S}(x, y')^{\alpha}) \overline{q}(x, y'),
\end{aligned}
$$

where we denote by $\overline{q}(x, y') = \sum_{y \in \mathcal{Y}} p(x, y) \overline{\ell}(y', y) \in [0, 1]$ and $\mathcal{S}(x, y) = \frac{e^{h(x, y)}}{\sum_{y'' \in \mathcal{Y}} e^{h(x, y'')}} \in [0, 1]$ with the constraint that $\sum_{y \in \mathcal{Y}} \mathcal{S}(x, y) = 1$. Let $y_{\min} = \operatorname{argmin}_{y' \in \mathcal{Y}} \sum_{y \in \mathcal{Y}} p(x, y) \ell(y', y)$, where we choose the label with the highest index under the natural ordering of labels as the tie-breaking strategy. For any $h \in \mathcal{H}$ such that $\mathsf{h}(x) \neq y_{\min}$ and $x \in \mathcal{X}$, by the symmetry and completeness of $\mathcal{H}$, we can always find a family of hypotheses $\{\overline{h}_{\mu} : \mu \in [-\mathcal{S}(x, y_{\min}), \mathcal{S}(x, \mathsf{h}(x))]\} \subset \mathcal{H}$ such that $\overline{\mathcal{S}}_{\mu}(x, \cdot) = \frac{e^{h_{\mu}(x, \cdot)}}{\sum_{y' \in \mathcal{Y}} e^{h_{\mu}(x, y')}}$ take the following values:

$$
\overline{\mathcal{S}}_{\mu}(x, y) = \begin{cases} \mathcal{S}(x, y) & \text{if } y \notin \{y_{\min}, \mathsf{h}(x)\} \\ \mathcal{S}(x, y_{\min}) + \mu & \text{if } y = \mathsf{h}(x) \\ \mathcal{S}(x, \mathsf{h}(x)) - \mu & \text{if } y = y_{\min}. \end{cases} \tag{24}
$$

Note that $\overline{\mathcal{S}}_{\mu}$ satisfies the constraint:

$$
\sum_{y \in \mathcal{Y}} \overline{\mathcal{S}}_{\mu}(x, y) = \sum_{y \in \mathcal{Y}} \mathcal{S}(x, y) = 1, \ \forall \mu \in [-\mathcal{S}(x, y_{\min}), \mathcal{S}(x, \mathsf{h}(x))].
$$

By (24) and using the fact that $\mathsf{H}(x) = \mathcal{Y}$ when $\mathcal{H}$ is symmetric, we obtain

$$
\begin{aligned}
&\Delta \mathfrak{C}_{\mathsf{L}^{\mathrm{comp}}, \mathcal{H}}(h, x) \\
&= \mathfrak{C}_{\mathsf{L}^{\mathrm{comp}}}(h, x) - \mathfrak{C}^{*}_{\mathsf{L}^{\mathrm{comp}}}(\mathcal{H}, x) \\
&\geq \mathfrak{C}_{\mathsf{L}^{\mathrm{comp}}}(h, x) - \inf_{\mu \in \mathbb{R}} \mathfrak{C}_{\mathsf{L}^{\mathrm{comp}}}(\overline{h}_{\mu}, x) \\
&= \frac{1}{\alpha} \sup_{\mu \in [-\mathcal{S}(x, y_{\min}), \mathcal{S}(x, \mathsf{h}(x))]} \left\{ \overline{q}(x, y_{\min}) \left[ -\mathcal{S}(x, y_{\min})^{\alpha} + (\mathcal{S}(x, \mathsf{h}(x)) - \mu)^{\alpha} \right] \right. \\
&\qquad \left. + \overline{q}(x, \mathsf{h}(x)) \left[ -\mathcal{S}(x, \mathsf{h}(x))^{\alpha} + (\mathcal{S}(x, y_{\min}) + \mu)^{\alpha} \right] \right\} \\
&= \frac{1}{\alpha} (\mathcal{S}(x, \mathsf{h}(x)) + \mathcal{S}(x, y_{\min}))^{\alpha} \left( \overline{q}(x, y_{\min})^{\frac{1}{1-\alpha}} + \overline{q}(x, \mathsf{h}(x))^{\frac{1}{1-\alpha}} \right)^{1-\alpha} \\
&\qquad - \frac{1}{\alpha} \overline{q}(x, y_{\min}) \mathcal{S}(x, y_{\min})^{\alpha} - \frac{1}{\alpha} \overline{q}(x, \mathsf{h}(x)) \mathcal{S}(x, \mathsf{h}(x))^{\alpha}
\end{aligned}
$$

(differentiating with respect to $\mu$ to optimize, optimum $\mu^{*} = \frac{\overline{q}(x, \mathsf{h}(x))^{\frac{1}{1-\alpha}} \mathcal{S}(x, \mathsf{h}(x)) - \overline{q}(x, y_{\min})^{\frac{1}{1-\alpha}} \mathcal{S}(x, y_{\min})}{\overline{q}(x, y_{\min})^{\frac{1}{1-\alpha}} + \overline{q}(x, \mathsf{h}(x))^{\frac{1}{1-\alpha}}}$)

$$
\geq \frac{1}{\alpha n^{\alpha}} \left[ 2^{\alpha} \left( \overline{q}(x, y_{\min})^{\frac{1}{1-\alpha}} + \overline{q}(x, \mathsf{h}(x))^{\frac{1}{1-\alpha}} \right)^{1-\alpha} - \overline{q}(x, y_{\min}) - \overline{q}(x, \mathsf{h}(x)) \right]
$$

(differentiating with respect to $\mathcal{S}$ to minimize, minimum is attained when $\mathcal{S}(x, \mathsf{h}(x)) = \mathcal{S}(x, y_{\min}) = \frac{1}{n}$)

$$
\begin{aligned}
&\geq \frac{(\overline{q}(x, \mathsf{h}(x)) - \overline{q}(x, y_{\min}))^{2}}{4n^{\alpha}} \quad \left( \left( \frac{a^{\frac{1}{1-\alpha}} + b^{\frac{1}{1-\alpha}}}{2} \right)^{1-\alpha} - \frac{a+b}{2} \geq \frac{\alpha}{4}(a-b)^{2}, \forall a, b \in [0, 1], 0 \leq a + b \leq 1 \right) \\
&= \frac{\left( \sum_{y \in \mathcal{Y}} p(x, y) \ell(\mathsf{h}(x), y) - \sum_{y \in \mathcal{Y}} p(x, y) \ell(y_{\min}, y) \right)^{2}}{4n^{\alpha}} \\
&= \frac{\Delta \mathfrak{C}_{\mathsf{L}, \mathcal{H}}(h, x)^{2}}{4n^{\alpha}}. \hspace{4cm} \text{(by Lemma 3 and } \mathsf{H}(x) = \mathcal{Y})
\end{aligned}
$$

Since the function $t \mapsto \frac{t^2}{4n^\alpha}$ is convex, by Jensen's inequality, we obtain for any hypothesis $h \in \mathcal{H}$ and any distribution,

$$
\begin{aligned}
\frac{\left(\mathcal{R}_\mathsf{L}(h) - \mathcal{R}^*_{\mathsf{L},\mathcal{H}} + \mathcal{M}_{\mathsf{L},\mathcal{H}}\right)^2}{4n^\alpha} &= \frac{\left(\mathbb{E}_X[\Delta\mathcal{C}_{\mathsf{L},\mathcal{H}}(h,x)]\right)^2}{4n^\alpha} \\
&\le \mathbb{E}_X\left[\frac{\Delta\mathcal{C}_{\mathsf{L},\mathcal{H}}(h,x)^2}{4n^\alpha}\right] \\
&\le \mathbb{E}_X[\Delta\mathcal{C}_{\mathsf{L}^{\mathrm{comp}},\mathcal{H}}(h,x)] \\
&= \mathcal{R}_{\mathsf{L}^{\mathrm{comp}}}(h) - \mathcal{R}^*_{\mathsf{L}^{\mathrm{comp}},\mathcal{H}} + \mathcal{M}_{\mathsf{L}^{\mathrm{comp}},\mathcal{H}},
\end{aligned}
$$

which leads to

$$
\mathcal{R}_\mathsf{L}(h) - \mathcal{R}^*_{\mathsf{L},\mathcal{H}} \le 2n^{\frac{\alpha}{2}}\left(\mathcal{R}_{\mathsf{L}^{\mathrm{comp}}_{\mathrm{gce}}}(h) - \mathcal{R}^*_{\mathsf{L}^{\mathrm{comp}}_{\mathrm{gce}},\mathcal{H}} + \mathcal{M}_{\mathsf{L}^{\mathrm{comp}}_{\mathrm{gce}},\mathcal{H}}\right)^{\frac{1}{2}} - \mathcal{M}_{\mathsf{L},\mathcal{H}}.
$$

$\square$

### E.4 Structured mean absolute error loss

**Theorem 14** ($\mathcal{H}$-**consistency bound of** $\mathsf{L}^{\mathrm{comp}}_{\mathrm{mae}}$)**.** *Assume that $\mathcal{H}$ is symmetric and complete. Then, for any target loss $\ell$, hypothesis $h \in \mathcal{H}$ and any distribution,*

$$
\mathcal{R}_\mathsf{L}(h) - \mathcal{R}^*_{\mathsf{L},\mathcal{H}} \le n\left(\mathcal{R}_{\mathsf{L}^{\mathrm{comp}}_{\mathrm{mae}}}(h) - \mathcal{R}^*_{\mathsf{L}^{\mathrm{comp}}_{\mathrm{mae}},\mathcal{H}} + \mathcal{M}_{\mathsf{L}^{\mathrm{comp}}_{\mathrm{mae}},\mathcal{H}}\right) - \mathcal{M}_{\mathsf{L},\mathcal{H}}. \tag{25}
$$

*Proof.* For the comp-sum structured loss $\mathsf{L}^{\mathrm{comp}}_{\mathrm{mae}}$, the conditional $\mathsf{L}^{\mathrm{comp}}_{\mathrm{mae}}$-risk can be expressed as follows:

$$
\begin{aligned}
\mathcal{C}_{\mathsf{L}^{\mathrm{comp}}_{\mathrm{mae}}}(h,x) &= \sum_{y \in \mathcal{Y}} p(x,y) \sum_{y' \in \mathcal{Y}} \overline{\ell}(y',y)\left(1 - \frac{e^{h(x,y')}}{\sum_{y'' \in \mathcal{Y}} e^{h(x,y'')}}\right) \\
&= \sum_{y' \in \mathcal{Y}} \left(1 - \frac{e^{h(x,y')}}{\sum_{y'' \in \mathcal{Y}} e^{h(x,y'')}}\right) \sum_{y \in \mathcal{Y}} p(x,y)\overline{\ell}(y',y) \\
&= \sum_{y' \in \mathcal{Y}} (1 - \mathcal{S}(x,y'))\overline{q}(x,y'),
\end{aligned}
$$

where we denote by $\overline{q}(x,y') = \sum_{y \in \mathcal{Y}} p(x,y)\overline{\ell}(y',y) \in [0,1]$ and $\mathcal{S}(x,y) = \frac{e^{h(x,y)}}{\sum_{y'' \in \mathcal{Y}} e^{h(x,y'')}} \in [0,1]$ with the constraint that $\sum_{y \in \mathcal{Y}} \mathcal{S}(x,y) = 1$. Let $y_{\min} = \operatorname{argmin}_{y' \in \mathcal{Y}} \sum_{y \in \mathcal{Y}} p(x,y)\ell(y',y)$, where we choose the label with the highest index under the natural ordering of labels as the tie-breaking strategy. For any $h \in \mathcal{H}$ such that $\mathsf{h}(x) \ne y_{\min}$ and $x \in \mathcal{X}$, by the symmetry and completeness of $\mathcal{H}$, we can always find a family of hypotheses $\left\{\overline{h}_\mu : \mu \in [-\mathcal{S}(x,y_{\min}), \mathcal{S}(x,\mathsf{h}(x))]\right\} \subset \mathcal{H}$ such that $\overline{\mathcal{S}}_\mu(x,\cdot) = \frac{e^{h_\mu(x,\cdot)}}{\sum_{y' \in \mathcal{Y}} e^{h_\mu(x,y')}}$ take the following values:

$$
\overline{\mathcal{S}}_\mu(x,y) = \begin{cases} \mathcal{S}(x,y) & \text{if } y \notin \{y_{\min}, \mathsf{h}(x)\} \\ \mathcal{S}(x,y_{\min}) + \mu & \text{if } y = \mathsf{h}(x) \\ \mathcal{S}(x,\mathsf{h}(x)) - \mu & \text{if } y = y_{\min}. \end{cases} \tag{26}
$$

Note that $\overline{\mathcal{S}}_\mu$ satisfies the constraint:

$$
\sum_{y \in \mathcal{Y}} \overline{\mathcal{S}}_\mu(x,y) = \sum_{y \in \mathcal{Y}} \mathcal{S}(x,y) = 1, \ \forall \mu \in [-\mathcal{S}(x,y_{\min}), \mathcal{S}(x,\mathsf{h}(x))].
$$

By (26) and using the fact that $H(x) = \mathcal{Y}$ when $\mathcal{H}$ is symmetric, we obtain

$$
\begin{aligned}
&\Delta\mathcal{C}_{\mathsf{L}^{\mathrm{comp}},\mathcal{H}}(h,x)\\
&= \mathcal{C}_{\mathsf{L}^{\mathrm{comp}}}(h,x) - \mathcal{C}^*_{\mathsf{L}^{\mathrm{comp}}}(\mathcal{H},x)\\
&\geq \mathcal{C}_{\mathsf{L}^{\mathrm{comp}}}(h,x) - \inf_{\mu\in\mathbb{R}}\mathcal{C}_{\mathsf{L}^{\mathrm{comp}}}(\overline{h}_\mu,x)\\
&= \sup_{\mu\in[-\mathcal{S}(x,y_{\min}),\mathcal{S}(x,h(x))]}\Bigg\{\overline{q}(x,y_{\min})\big[-\mathcal{S}(x,y_{\min}) + \mathcal{S}(x,h(x)) - \mu\big]\\
&\qquad\qquad + \overline{q}(x,h(x))\big[-\mathcal{S}(x,h(x)) + \mathcal{S}(x,y_{\min}) + \mu\big]\Bigg\}\\
&= \overline{q}(x,y_{\min})\mathcal{S}(x,h(x)) - \overline{q}(x,h(x))\mathcal{S}(x,h(x))\\
&\qquad\quad (\text{differentiating with respect to } \mu \text{ to optimize, optimum } \mu^* = -\mathcal{S}(x,y_{\min}))\\
&\geq \frac{1}{n}\big(\overline{q}(x,y_{\min}) - \overline{q}(x,h(x))\big)\\
&\qquad (\text{differentiating with respect to } \mathcal{S} \text{ to minimize, minimum is attained when } \mathcal{S}(x,h(x)) = \tfrac{1}{n})\\
&= \frac{\sum_{y\in\mathcal{Y}}p(x,y)\ell(h(x),y) - \sum_{y\in\mathcal{Y}}p(x,y)\ell(y_{\min},y)}{n}\\
&= \frac{\Delta\mathcal{C}_{\mathsf{L},\mathcal{H}}(h,x)}{n} \qquad\qquad\qquad\qquad\qquad (\text{by Lemma 3 and } H(x) = \mathcal{Y})
\end{aligned}
$$

Therefore, we obtain for any hypothesis $h\in\mathcal{H}$ and any distribution,

$$
\begin{aligned}
\frac{\mathcal{R}_{\mathsf{L}}(h) - \mathcal{R}^*_{\mathsf{L},\mathcal{H}} + \mathcal{M}_{\mathsf{L},\mathcal{H}}}{n} &= \frac{\mathbb{E}_X[\Delta\mathcal{C}_{\mathsf{L},\mathcal{H}}(h,x)]}{n}\\
&= \mathbb{E}_X[\Delta\mathcal{C}_{\mathsf{L}^{\mathrm{comp}},\mathcal{H}}(h,x)]\\
&= \mathcal{R}_{\mathsf{L}^{\mathrm{comp}}}(h) - \mathcal{R}^*_{\mathsf{L}^{\mathrm{comp}},\mathcal{H}} + \mathcal{M}_{\mathsf{L}^{\mathrm{comp}},\mathcal{H}},
\end{aligned}
$$

which leads to

$$
\mathcal{R}_{\mathsf{L}}(h) - \mathcal{R}^*_{\mathsf{L},\mathcal{H}} \leq n\big(\mathcal{R}_{\mathsf{L}_{\mathrm{mae}}^{\mathrm{comp}}}(h) - \mathcal{R}^*_{\mathsf{L}_{\mathrm{mae}}^{\mathrm{comp}},\mathcal{H}} + \mathcal{M}_{\mathsf{L}_{\mathrm{mae}}^{\mathrm{comp}},\mathcal{H}}\big) - \mathcal{M}_{\mathsf{L},\mathcal{H}}.
$$

$\qquad\square$

## F  Proofs for structured constrained losses

### F.1  Structured constrained exponential loss

**Theorem 15** ($\mathcal{H}$-consistency bound of $\mathsf{L}_{\exp}^{\mathrm{cstnd}}$)**.** *Assume that $\mathcal{H}$ is symmetric and complete. Then, for any target loss $\ell$, hypothesis $h\in\mathcal{H}$ and any distribution,*

$$
\mathcal{R}_{\mathsf{L}}(h) - \mathcal{R}^*_{\mathsf{L},\mathcal{H}} \leq 2\sqrt{\ell_{\max}}\Big(\mathcal{R}_{\mathsf{L}_{\exp}^{\mathrm{cstnd}}}(h) - \mathcal{R}^*_{\mathsf{L}_{\exp}^{\mathrm{cstnd}},\mathcal{H}} + \mathcal{M}_{\mathsf{L}_{\exp}^{\mathrm{cstnd}},\mathcal{H}}\Big)^{\frac{1}{2}} - \mathcal{M}_{\mathsf{L},\mathcal{H}}. \tag{27}
$$

*Proof.* Denote by $q(x,y') = \sum_{y\in\mathcal{Y}}p(x,y)\ell(y',y) \in [0,\ell_{\max}]$. For the constrained structured loss $\mathsf{L}_{\exp}^{\mathrm{cstnd}}$, the conditional $\mathsf{L}_{\exp}^{\mathrm{cstnd}}$-risk can be expressed as follows:

$$
\mathcal{C}_{\mathsf{L}_{\exp}^{\mathrm{cstnd}}}(h,x) = \sum_{y\in\mathcal{Y}}p(x,y)\sum_{y'\in\mathcal{Y}}\ell(y,y')e^{h(x,y')} = \sum_{y'\in\mathcal{Y}}e^{h(x,y')}q(x,y').
$$

Let $y_{\min} = \operatorname{argmin}_{y'\in\mathcal{Y}}\sum_{y\in\mathcal{Y}}p(x,y)\ell(y',y)$, where we choose the label with the highest index under the natural ordering of labels as the tie-breaking strategy. For any $h\in\mathcal{H}$ such that $h(x)\neq y_{\min}$ and $x\in\mathcal{X}$, by the symmetry and completeness of $\mathcal{H}$, we can always find a family of hypotheses $\{\overline{h}_\mu : \mu\in\mathbb{R}\} \subset \mathcal{H}$ such that $h_\mu(x,\cdot)$ take the following values:

$$
\overline{h}_\mu(x,y) = \begin{cases} h(x,y) & \text{if } y\notin\{y_{\min},h(x)\}\\ h(x,y_{\min}) + \mu & \text{if } y = h(x)\\ h(x,h(x)) - \mu & \text{if } y = y_{\min}. \end{cases} \tag{28}
$$

Note that the hypotheses $\overline{h}_\mu$ satisfies the constraint:

$$\sum_{y\in\mathcal{Y}} \overline{h}_\mu(x,y) = \sum_{y\in\mathcal{Y}} h(x,y) = 0, \ \forall\mu\in\mathbb{R}.$$

Since $\sum_{y\in\mathcal{Y}} h(x,y) = 0$, there must be non-negative scores. By definition of $\mathsf{h}(x)$ as a maximizer, we must thus have $h(x,\mathsf{h}(x)) \geq 0$. By (28) and using the fact that $\mathsf{H}(x) = \mathcal{Y}$ when $\mathcal{H}$ is symmetric, we obtain

$$
\begin{aligned}
&\Delta\mathcal{C}_{\mathsf{L}^{\mathrm{cstnd}},\mathcal{H}}(h,x) \\
&= \mathcal{C}_{\mathsf{L}^{\mathrm{cstnd}}}(h,x) - \mathcal{C}^*_{\mathsf{L}^{\mathrm{cstnd}}}(\mathcal{H},x) \\
&\geq \mathcal{C}_{\mathsf{L}^{\mathrm{cstnd}}}(h,x) - \inf_{\mu\in\mathbb{R}} \mathcal{C}_{\mathsf{L}^{\mathrm{cstnd}}}(\overline{h}_\mu,x) \\
&= \sup_{\mu\in\mathbb{R}}\Big\{ q(x,y_{\min})\big(e^{h(x,y_{\min})} - e^{h(x,\mathsf{h}(x))-\mu}\big) + q(x,\mathsf{h}(x))\big(e^{h(x,\mathsf{h}(x))} - e^{h(x,y_{\min})+\mu}\big)\Big\} \\
&= \left(\sqrt{q(x,\mathsf{h}(x))e^{h(x,\mathsf{h}(x))}} - \sqrt{q(x,y_{\min})e^{h(x,y_{\min})}}\right)^2 \\
&\qquad\text{(differentiating with respect to $\mu$ to optimize, optimum $\mu^* = \frac{1}{2}\log\frac{q(x,y_{\min})e^{h(x,\mathsf{h}(x))}}{q(x,\mathsf{h}(x))e^{h(x,y_{\min})}}$)} \\
&\geq e^{h(x,\mathsf{h}(x))}\left(\sqrt{q(x,y_{\min})} - \sqrt{q(x,\mathsf{h}(x))}\right)^2 \\
&\qquad\qquad\qquad (e^{h(x,\mathsf{h}(x))} \geq e^{h(x,y_{\min})} \text{ and } q(x,\mathsf{h}(x)) \geq q(x,y_{\min})) \\
&\geq \left(\sqrt{q(x,y_{\min})} - \sqrt{q(x,\mathsf{h}(x))}\right)^2 \qquad\qquad\qquad\qquad (h(x,\mathsf{h}(x)) \geq 0) \\
&= \left(\frac{q(x,\mathsf{h}(x)) - q(x,y_{\min})}{\sqrt{q(x,y_{\min})} + \sqrt{q(x,\mathsf{h}(x))}}\right)^2 \\
&\geq \frac{1}{4\ell_{\max}}\big(q(x,\mathsf{h}(x)) - q(x,y_{\min})\big)^2 \qquad\qquad\qquad (0 \leq q(x,y) \leq \ell_{\max}) \\
&= \frac{1}{4\ell_{\max}}\Delta\mathcal{C}_{\mathsf{L},\mathcal{H}}(h,x)^2. \qquad\qquad\qquad\text{(by Lemma 3 and } \mathsf{H}(x) = \mathcal{Y})
\end{aligned}
$$

Since the function $t \mapsto \frac{t^2}{4\ell_{\max}}$ is convex, by Jensen's inequality, we obtain for any hypothesis $h \in \mathcal{H}$ and any distribution,

$$
\begin{aligned}
\frac{\big(\mathcal{R}_{\mathsf{L}}(h) - \mathcal{R}^*_{\mathsf{L},\mathcal{H}} + \mathcal{M}_{\mathsf{L},\mathcal{H}}\big)^2}{4\ell_{\max}} &= \frac{\big(\mathbb{E}_X[\Delta\mathcal{C}_{\mathsf{L},\mathcal{H}}(h,x)]\big)^2}{4\ell_{\max}} \\
&\leq \mathbb{E}_X\left[\frac{\Delta\mathcal{C}_{\mathsf{L},\mathcal{H}}(h,x)^2}{4\ell_{\max}}\right] \\
&\leq \mathbb{E}_X\big[\Delta\mathcal{C}_{\mathsf{L}^{\mathrm{cstnd}},\mathcal{H}}(h,x)\big] \\
&= \mathcal{R}_{\mathsf{L}^{\mathrm{cstnd}}}(h) - \mathcal{R}^*_{\mathsf{L}^{\mathrm{cstnd}},\mathcal{H}} + \mathcal{M}_{\mathsf{L}^{\mathrm{cstnd}},\mathcal{H}},
\end{aligned}
$$

which leads to

$$\mathcal{R}_{\mathsf{L}}(h) - \mathcal{R}^*_{\mathsf{L},\mathcal{H}} \leq 2\sqrt{\ell_{\max}}\Big(\mathcal{R}_{\mathsf{L}^{\mathrm{cstnd}}_{\exp}}(h) - \mathcal{R}^*_{\mathsf{L}^{\mathrm{cstnd}}_{\exp},\mathcal{H}} + \mathcal{M}_{\mathsf{L}^{\mathrm{cstnd}}_{\exp},\mathcal{H}}\Big)^{\frac{1}{2}} - \mathcal{M}_{\mathsf{L},\mathcal{H}}.$$

$\square$

### F.2 Structured constrained squared-hinge loss

**Theorem 16** ($\mathcal{H}$-consistency bound of $\mathsf{L}^{\mathrm{cstnd}}_{\mathrm{sq-hinge}}$)**.** *Assume that $\mathcal{H}$ is symmetric and complete. Then, for any target loss $\ell$, hypothesis $h \in \mathcal{H}$ and any distribution,*

$$\mathcal{R}_{\mathsf{L}}(h) - \mathcal{R}^*_{\mathsf{L},\mathcal{H}} \leq \Big(\mathcal{R}_{\mathsf{L}^{\mathrm{cstnd}}_{\mathrm{sq-hinge}}}(h) - \mathcal{R}^*_{\mathsf{L}^{\mathrm{cstnd}}_{\mathrm{sq-hinge}},\mathcal{H}} + \mathcal{M}_{\mathsf{L}^{\mathrm{cstnd}}_{\mathrm{sq-hinge}},\mathcal{H}}\Big)^{\frac{1}{2}} - \mathcal{M}_{\mathsf{L},\mathcal{H}}. \tag{29}$$

*Proof.* Denote by $q(x, y') = \sum_{y \in \mathcal{Y}} p(x, y) \ell(y', y) \in [0, \ell_{\max}]$. For the constrained structured loss $\mathsf{L}_{\text{sq-hinge}}^{\text{cstnd}}$, the conditional $\mathsf{L}_{\text{sq-hinge}}^{\text{cstnd}}$-risk can be expressed as follows:

$$\mathcal{C}_{\mathsf{L}_{\text{sq-hinge}}^{\text{cstnd}}}(h, x) = \sum_{y \in \mathcal{Y}} p(x, y) \sum_{y' \in \mathcal{Y}} \ell(y', y) \max\{0, 1 + h(x, y')\}^2$$

$$= \sum_{y' \in \mathcal{Y}} \max\{0, 1 + h(x, y')\}^2 q(x, y').$$

Let $y_{\min} = \arg\min_{y' \in \mathcal{Y}} \sum_{y \in \mathcal{Y}} p(x, y) \ell(y', y)$, where we choose the label with the highest index under the natural ordering of labels as the tie-breaking strategy. For any $h \in \mathcal{H}$ such that $\mathsf{h}(x) \neq y_{\min}$ and $x \in \mathcal{X}$, by the symmetry and completeness of $\mathcal{H}$, we can always find a family of hypotheses $\{\overline{h}_\mu : \mu \in \mathbb{R}\} \subset \mathcal{H}$ such that $h_\mu(x, \cdot)$ take the following values:

$$\overline{h}_\mu(x, y) = \begin{cases} h(x, y) & \text{if } y \notin \{y_{\min}, \mathsf{h}(x)\} \\ h(x, y_{\min}) + \mu & \text{if } y = \mathsf{h}(x) \\ h(x, \mathsf{h}(x)) - \mu & \text{if } y = y_{\min}. \end{cases} \tag{30}$$

Note that the hypotheses $\overline{h}_\mu$ satisfies the constraint:

$$\sum_{y \in \mathcal{Y}} \overline{h}_\mu(x, y) = \sum_{y \in \mathcal{Y}} h(x, y) = 0, \ \forall \mu \in \mathbb{R}.$$

Since $\sum_{y \in \mathcal{Y}} h(x, y) = 0$, there must be non-negative scores. By definition of $\mathsf{h}(x)$ as a maximizer, we must thus have $h(x, \mathsf{h}(x)) \geq 0$. By (30) and using the fact that $\mathsf{H}(x) = \mathcal{Y}$ when $\mathcal{H}$ is symmetric, we obtain

$$\Delta \mathcal{C}_{\mathsf{L}^{\text{cstnd}}, \mathcal{H}}(h, x)$$

$$= \mathcal{C}_{\mathsf{L}^{\text{cstnd}}}(h, x) - \mathcal{C}^*_{\mathsf{L}^{\text{cstnd}}}(\mathcal{H}, x)$$

$$\geq \mathcal{C}_{\mathsf{L}^{\text{cstnd}}}(h, x) - \inf_{\mu \in \mathbb{R}} \mathcal{C}_{\mathsf{L}^{\text{cstnd}}}(\overline{h}_\mu, x)$$

$$= \sup_{\mu \in \mathbb{R}} \Big\{ q(x, y_{\min}) \Big( \max\{0, 1 + h(x, y_{\min})\}^2 - \max\{0, 1 + h(x, \mathsf{h}(x)) - \mu\}^2 \Big)$$

$$+ q(x, \mathsf{h}(x)) \Big( \max\{0, 1 + h(x, \mathsf{h}(x))\}^2 - \max\{0, 1 + h(x, y_{\min}) + \mu\}^2 \Big) \Big\}$$

$$\geq (1 + h(x, \mathsf{h}(x)))^2 (q(x, y_{\min}) - q(x, \mathsf{h}(x)))^2 \quad \text{(differentiating with respect to } \mu \text{ to optimize)}$$

$$\geq (q(x, \mathsf{h}(x)) - q(x, y_{\min}))^2 \qquad\qquad (h(x, \mathsf{h}(x)) \geq 0)$$

$$= \Big( \sum_{y \in \mathcal{Y}} p(x, y) \ell(\mathsf{h}(x), y) - \sum_{y \in \mathcal{Y}} p(x, y) \ell(y_{\min}, y) \Big)^2$$

$$= \Delta \mathcal{C}_{\mathsf{L}, \mathcal{H}}(h, x)^2. \qquad\qquad \text{(by Lemma 3 and } \mathsf{H}(x) = \mathcal{Y})$$

Since the function $t \mapsto t^2$ is convex, by Jensen's inequality, we obtain for any hypothesis $h \in \mathcal{H}$ and any distribution,

$$\Big( \mathcal{R}_{\mathsf{L}}(h) - \mathcal{R}^*_{\mathsf{L}, \mathcal{H}} + \mathcal{M}_{\mathsf{L}, \mathcal{H}} \Big)^2 = \Big( \mathbb{E}_X [\Delta \mathcal{C}_{\mathsf{L}, \mathcal{H}}(h, x)] \Big)^2$$

$$\leq \mathbb{E}_X \big[ \Delta \mathcal{C}_{\mathsf{L}, \mathcal{H}}(h, x)^2 \big]$$

$$\leq \mathbb{E}_X \big[ \Delta \mathcal{C}_{\mathsf{L}^{\text{cstnd}}, \mathcal{H}}(h, x) \big]$$

$$= \mathcal{R}_{\mathsf{L}^{\text{cstnd}}}(h) - \mathcal{R}^*_{\mathsf{L}^{\text{cstnd}}, \mathcal{H}} + \mathcal{M}_{\mathsf{L}^{\text{cstnd}}, \mathcal{H}},$$

which leads to

$$\mathcal{R}_{\mathsf{L}}(h) - \mathcal{R}^*_{\mathsf{L}, \mathcal{H}} \leq \Big( \mathcal{R}_{\mathsf{L}_{\text{sq-hinge}}^{\text{cstnd}}}(h) - \mathcal{R}^*_{\mathsf{L}_{\text{sq-hinge}}^{\text{cstnd}}, \mathcal{H}} + \mathcal{M}_{\mathsf{L}_{\text{sq-hinge}}^{\text{cstnd}}, \mathcal{H}} \Big)^{\frac{1}{2}} - \mathcal{M}_{\mathsf{L}, \mathcal{H}}.$$

$\square$

## F.3 Structured constrained hinge loss

**Theorem 17** ($\mathcal{H}$-**consistency bound of** $\mathsf{L}_{\text{hinge}}^{\text{cstnd}}$). *Assume that $\mathcal{H}$ is symmetric and complete. Then, for any target loss $\ell$, hypothesis $h \in \mathcal{H}$ and any distribution,*

$$\mathcal{R}_{\mathsf{L}}(h) - \mathcal{R}_{\mathsf{L},\mathcal{H}}^* \le \mathcal{R}_{\mathsf{L}_{\text{hinge}}^{\text{cstnd}}}(h) - \mathcal{R}_{\mathsf{L}_{\text{hinge}}^{\text{cstnd}},\mathcal{H}}^* + \mathcal{M}_{\mathsf{L}_{\text{hinge}}^{\text{cstnd}},\mathcal{H}} - \mathcal{M}_{\mathsf{L},\mathcal{H}}. \tag{31}$$

*Proof.* Denote by $q(x,y') = \sum_{y \in \mathcal{Y}} p(x,y)\ell(y',y) \in [0, \ell_{\max}]$. For the constrained structured loss $\mathsf{L}_{\text{hinge}}^{\text{cstnd}}$, the conditional $\mathsf{L}_{\text{hinge}}^{\text{cstnd}}$-risk can be expressed as follows:

$$\begin{aligned}
\mathcal{C}_{\mathsf{L}_{\text{hinge}}^{\text{cstnd}}}(h,x) &= \sum_{y \in \mathcal{Y}} p(x,y) \sum_{y' \in \mathcal{Y}} \ell(y',y) \max\{0, 1 + h(x,y')\} \\
&= \sum_{y' \in \mathcal{Y}} \max\{0, 1 + h(x,y')\} q(x,y').
\end{aligned}$$

Let $y_{\min} = \operatorname{argmin}_{y' \in \mathcal{Y}} \sum_{y \in \mathcal{Y}} p(x,y)\ell(y',y)$, where we choose the label with the highest index under the natural ordering of labels as the tie-breaking strategy. For any $h \in \mathcal{H}$ such that $\mathsf{h}(x) \ne y_{\min}$ and $x \in \mathcal{X}$, by the symmetry and completeness of $\mathcal{H}$, we can always find a family of hypotheses $\{\overline{h}_\mu : \mu \in \mathbb{R}\} \subset \mathcal{H}$ such that $h_\mu(x,\cdot)$ take the following values:

$$\overline{h}_\mu(x,y) = \begin{cases} h(x,y) & \text{if } y \notin \{y_{\min}, \mathsf{h}(x)\} \\ h(x,y_{\min}) + \mu & \text{if } y = \mathsf{h}(x) \\ h(x,\mathsf{h}(x)) - \mu & \text{if } y = y_{\min}. \end{cases} \tag{32}$$

Note that the hypotheses $\overline{h}_\mu$ satisfies the constraint:

$$\sum_{y \in \mathcal{Y}} \overline{h}_\mu(x,y) = \sum_{y \in \mathcal{Y}} h(x,y) = 0, \ \forall \mu \in \mathbb{R}.$$

Since $\sum_{y \in \mathcal{Y}} h(x,y) = 0$, there must be non-negative scores. By definition of $\mathsf{h}(x)$ as a maximizer, we must thus have $h(x,\mathsf{h}(x)) \ge 0$. By (32) and using the fact that $\mathsf{H}(x) = \mathcal{Y}$ when $\mathcal{H}$ is symmetric, we obtain

$$\begin{aligned}
&\Delta\mathcal{C}_{\mathsf{L}^{\text{cstnd}},\mathcal{H}}(h,x) \\
&= \mathcal{C}_{\mathsf{L}^{\text{cstnd}}}(h,x) - \mathcal{C}_{\mathsf{L}^{\text{cstnd}}}^*(\mathcal{H},x) \\
&\ge \mathcal{C}_{\mathsf{L}^{\text{cstnd}}}(h,x) - \inf_{\mu \in \mathbb{R}} \mathcal{C}_{\mathsf{L}^{\text{cstnd}}}(\overline{h}_\mu,x) \\
&= \sup_{\mu \in \mathbb{R}} \Big\{ q(x,y_{\min})(\max\{0, 1 + h(x,y_{\min})\} - \max\{0, 1 + h(x,\mathsf{h}(x)) - \mu\}) \\
&\quad + q(x,\mathsf{h}(x))(\max\{0, 1 + h(x,\mathsf{h}(x))\} - \max\{0, 1 + h(x,y_{\min}) + \mu\}) \Big\} \\
&\ge (1 + h(x,\mathsf{h}(x)))(q(x,\mathsf{h}(x)) - q(x,y_{\min})) \quad \text{(differentiating with respect to } \mu \text{ to optimize)} \\
&\ge q(x,\mathsf{h}(x)) - q(x,y_{\min}) \quad\quad\quad\quad\quad\quad\quad\quad (h(x,\mathsf{h}(x)) \ge 0) \\
&= \sum_{y \in \mathcal{Y}} p(x,y)\ell(\mathsf{h}(x),y) - \sum_{y \in \mathcal{Y}} p(x,y)\ell(y_{\min},y) \\
&= \Delta\mathcal{C}_{\mathsf{L},\mathcal{H}}(h,x). \quad\quad\quad\quad\quad\quad\quad\quad\quad \text{(by Lemma 3 and } \mathsf{H}(x) = \mathcal{Y})
\end{aligned}$$

Therefore, we obtain for any hypothesis $h \in \mathcal{H}$ and any distribution,

$$\begin{aligned}
\mathcal{R}_{\mathsf{L}}(h) - \mathcal{R}_{\mathsf{L},\mathcal{H}}^* + \mathcal{M}_{\mathsf{L},\mathcal{H}} &= \mathbb{E}_X\big[\Delta\mathcal{C}_{\mathsf{L},\mathcal{H}}(h,x)\big] \\
&\le \mathbb{E}_X\big[\Delta\mathcal{C}_{\mathsf{L}^{\text{cstnd}},\mathcal{H}}(h,x)\big] \\
&= \mathcal{R}_{\mathsf{L}^{\text{cstnd}}}(h) - \mathcal{R}_{\mathsf{L}^{\text{cstnd}},\mathcal{H}}^* + \mathcal{M}_{\mathsf{L}^{\text{cstnd}},\mathcal{H}},
\end{aligned}$$

which leads to

$$\mathcal{R}_{\mathsf{L}}(h) - \mathcal{R}_{\mathsf{L},\mathcal{H}}^* \le \mathcal{R}_{\mathsf{L}_{\text{hinge}}^{\text{cstnd}}}(h) - \mathcal{R}_{\mathsf{L}_{\text{hinge}}^{\text{cstnd}},\mathcal{H}}^* + \mathcal{M}_{\mathsf{L}_{\text{hinge}}^{\text{cstnd}},\mathcal{H}} - \mathcal{M}_{\mathsf{L},\mathcal{H}}.$$

$\square$

### F.4 Structured constrained $\rho$-margin loss

**Theorem 18** ($\mathcal{H}$-**consistency bound of** $\mathsf{L}_\rho^{\mathrm{cstnd}}$). *Assume that $\mathcal{H}$ is symmetric and complete. Then, for any target loss $\ell$, hypothesis $h \in \mathcal{H}$ and any distribution,*

$$\mathcal{R}_\mathsf{L}(h) - \mathcal{R}_{\mathsf{L},\mathcal{H}}^* \le \mathcal{R}_{\mathsf{L}_\rho^{\mathrm{cstnd}}}(h) - \mathcal{R}_{\mathsf{L}_\rho^{\mathrm{cstnd}},\mathcal{H}}^* + \mathcal{M}_{\mathsf{L}_\rho^{\mathrm{cstnd}},\mathcal{H}} - \mathcal{M}_{\mathsf{L},\mathcal{H}}. \tag{33}$$

*Proof.* Denote by $q(x,y') = \sum_{y \in \mathcal{Y}} p(x,y)\ell(y',y) \in [0, \ell_{\max}]$. For the constrained structured loss $\mathsf{L}_\rho^{\mathrm{cstnd}}$, the conditional $\mathsf{L}_\rho^{\mathrm{cstnd}}$-risk can be expressed as follows:

$$\begin{aligned}
\mathcal{C}_{\mathsf{L}_\rho^{\mathrm{cstnd}}}(h,x) &= \sum_{y \in \mathcal{Y}} p(x,y) \sum_{y' \in \mathcal{Y}} \ell(y',y) \max\{0, 1 + h(x,y')\} \\
&= \sum_{y' \in \mathcal{Y}} \max\{0, 1 + h(x,y')\} q(x,y').
\end{aligned}$$

Let $y_{\min} = \operatorname{argmin}_{y' \in \mathcal{Y}} \sum_{y \in \mathcal{Y}} p(x,y)\ell(y',y)$, where we choose the label with the highest index under the natural ordering of labels as the tie-breaking strategy. For any $h \in \mathcal{H}$ such that $\mathsf{h}(x) \ne y_{\min}$ and $x \in \mathcal{X}$, by the symmetry and completeness of $\mathcal{H}$, we can always find a family of hypotheses $\{\overline{h}_\mu : \mu \in \mathbb{R}\} \subset \mathcal{H}$ such that $h_\mu(x,\cdot)$ take the following values:

$$\overline{h}_\mu(x,y) = \begin{cases} h(x,y) & \text{if } y \notin \{y_{\min}, \mathsf{h}(x)\} \\ h(x,y_{\min}) + \mu & \text{if } y = \mathsf{h}(x) \\ h(x,\mathsf{h}(x)) - \mu & \text{if } y = y_{\min}. \end{cases} \tag{34}$$

Note that the hypotheses $\overline{h}_\mu$ satisfies the constraint:

$$\sum_{y \in \mathcal{Y}} \overline{h}_\mu(x,y) = \sum_{y \in \mathcal{Y}} h(x,y) = 0, \ \forall \mu \in \mathbb{R}.$$

Since $\sum_{y \in \mathcal{Y}} h(x,y) = 0$, there must be non-negative scores. By definition of $\mathsf{h}(x)$ as a maximizer, we must thus have $h(x,\mathsf{h}(x)) \ge 0$. By (34) and using the fact that $\mathsf{H}(x) = \mathcal{Y}$ when $\mathcal{H}$ is symmetric, we obtain

$$\begin{aligned}
&\Delta\mathcal{C}_{\mathsf{L}^{\mathrm{cstnd}},\mathcal{H}}(h,x) \\
&= \mathcal{C}_{\mathsf{L}^{\mathrm{cstnd}}}(h,x) - \mathcal{C}_{\mathsf{L}^{\mathrm{cstnd}}}^*(\mathcal{H},x) \\
&\ge \mathcal{C}_{\mathsf{L}^{\mathrm{cstnd}}}(h,x) - \inf_{\mu \in \mathbb{R}} \mathcal{C}_{\mathsf{L}^{\mathrm{cstnd}}}(\overline{h}_\mu, x) \\
&= \sup_{\mu \in \mathbb{R}} \left\{ q(x,y_{\min})\left(\min\left\{\max\left\{0, 1 + \frac{h(x,y_{\min})}{\rho}\right\}, 1\right\} - \min\left\{\max\left\{0, 1 + \frac{h(x,\mathsf{h}(x)) - \mu}{\rho}\right\}, 1\right\}\right) \right. \\
&\quad \left. + q(x,\mathsf{h}(x))\left(\min\left\{\max\left\{0, 1 + \frac{h(x,\mathsf{h}(x))}{\rho}\right\}, 1\right\} - \min\left\{\max\left\{0, 1 + \frac{h(x,y_{\min}) + \mu}{\rho}\right\}, 1\right\}\right) \right\} \\
&\ge q(x,\mathsf{h}(x)) - q(x,y_{\min}) \qquad\qquad \text{(differentiating with respect to } \mu \text{ to optimize)} \\
&= \sum_{y \in \mathcal{Y}} p(x,y)\ell(\mathsf{h}(x),y) - \sum_{y \in \mathcal{Y}} p(x,y)\ell(y_{\min},y) \\
&= \Delta\mathcal{C}_{\mathsf{L},\mathcal{H}}(h,x). \qquad\qquad\qquad\qquad\qquad \text{(by Lemma 3 and } \mathsf{H}(x) = \mathcal{Y})
\end{aligned}$$

Therefore, we obtain for any hypothesis $h \in \mathcal{H}$ and any distribution,

$$\begin{aligned}
\mathcal{R}_\mathsf{L}(h) - \mathcal{R}_{\mathsf{L},\mathcal{H}}^* + \mathcal{M}_{\mathsf{L},\mathcal{H}} &= \mathbb{E}_X\big[\Delta\mathcal{C}_{\mathsf{L},\mathcal{H}}(h,x)\big] \\
&\le \mathbb{E}_X\big[\Delta\mathcal{C}_{\mathsf{L}^{\mathrm{cstnd}},\mathcal{H}}(h,x)\big] \\
&= \mathcal{R}_{\mathsf{L}^{\mathrm{cstnd}}}(h) - \mathcal{R}_{\mathsf{L}^{\mathrm{cstnd}},\mathcal{H}}^* + \mathcal{M}_{\mathsf{L}^{\mathrm{cstnd}},\mathcal{H}},
\end{aligned}$$

which leads to

$$\mathcal{R}_\mathsf{L}(h) - \mathcal{R}_{\mathsf{L},\mathcal{H}}^* \le \mathcal{R}_{\mathsf{L}_\rho^{\mathrm{cstnd}}}(h) - \mathcal{R}_{\mathsf{L}_\rho^{\mathrm{cstnd}},\mathcal{H}}^* + \mathcal{M}_{\mathsf{L}_\rho^{\mathrm{cstnd}},\mathcal{H}} - \mathcal{M}_{\mathsf{L},\mathcal{H}}.$$

$\square$

# G  Efficient gradient computation and inference

Here, we describe efficient algorithms for the computation of the gradients for the loss functions $\mathsf{L}_{\log}^{\text{comp}}$ and $\mathsf{L}_{\exp}^{\text{comp}}$. We also briefly discuss an efficient algorithm for inference.

## G.1  Efficient gradient computation for $\mathsf{L}_{\log}^{\text{comp}}$

We first present an efficient algorithm for the computation of the quantities $\mathsf{L}(\mathbf{z}, s)$ in the important case of rational losses, next in the case of Markovian losses.

**Rational losses.** *Rational losses* form a general family of loss functions based on rational kernels [Cortes et al., 2004] that includes, in particular, $n$-gram losses, which can be defined for a pair of sequences $(y, y')$ as the negative inner product of the vectors of $n$-gram counts of $y$ and $y'$.

Our algorithm bears some similarity to that of Cortes et al. [2018] for the computation of the gradient of the VCRF loss function. It is however distinct because the structured prediction loss function we are considering and our definition of rational loss are both different. We will adopt a similar notation and terminology. Recall that for any sequence $y$, we denote by $y_i$ the symbol in its $i$th position and by $y_i^j = y_i y_{i+1} \cdots y_j$ the substring of $y$ starting at position $i$ and ending at $j$. We denote by $\mathsf{E}_{\mathcal{A}}$ the set of transitions of a WFA $\mathcal{A}$.

Let $\mathcal{U}$ be a weighted finite-state transducer (WFST) over the $(+, \times)$ semiring over the reals, with $\Delta$ as both the input and output alphabet. Then, we define the rational loss associated to $\mathcal{U}$ for all $y, y' \in \Delta^*$ by $\overline{\ell}(y, y') = \mathcal{U}(y, y')$.

Let $\overline{\mathcal{Y}}$ denote a WFA over the $(+, \times)$ semiring accepting the set of all sequences of length $l$ with weight one and let $\mathcal{Y}_i$ denote the WFA accepting only $y_i$ with weight one. Then, by definition, the weighted transducer $\overline{\mathcal{Y}} \circ \mathcal{U} \circ \mathcal{Y}_i$ obtained by composition maps each sequence $y$ in $\Delta^l$ to $y_i$ with weight $\mathcal{U}(y, y_i)$. The WFA $\Pi_1(\overline{\mathcal{Y}} \circ \mathcal{U} \circ \mathcal{Y}_i)$ derived from that transducer by projection on the input (that is by removing output labels) is associating to each sequence $y$ weight $\mathcal{U}(y, y_i)$. We use weighted determinization [Mohri, 1997] to compute an equivalent deterministic WFA denote $\mathcal{M}$. As shown by Cortes et al. [2015][Theorem 3], $\mathcal{M}$ can be computed in polynomial time. $\mathcal{M}$

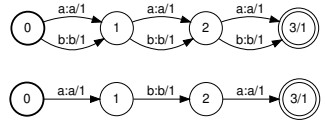

Figure 3: Illustration of the WFA $\overline{\mathcal{Y}}$ for $\Delta = \{a, b\}$ and $l = 3$, and the WFA $\mathcal{Y}_i$, where $y_i = aba$.

admits a unique path labeled with any sequence $y \in \Delta^l$ and the weight of that path is $\mathcal{U}(y, y_i)$. The weight of that accepting path is obtained by multiplying the weights of its transitions and that of the final state.

We now define a deterministic $p$-gram WFA $\mathcal{N}$ that accepts all sequences $y \in \Delta^l$ with each of its states $(\mathbf{z}', s)$ encoding a $(p-1)$-gram $\mathbf{z}'$ read to reach it and the position $s$ in the sequence $y$ at which it is reached. The transitions of $\mathcal{N}$ are therefore defined as follows with weight one:

$$\mathsf{E}_{\mathcal{N}} = \left\{ \left( (y_{s-p+1}^{s-1}, s-1), a, 1, (y_{s-p+2}^{s-1}a, s) \right) : y \in \Delta^l, a \in \Delta, s \in [l] \right\}.$$

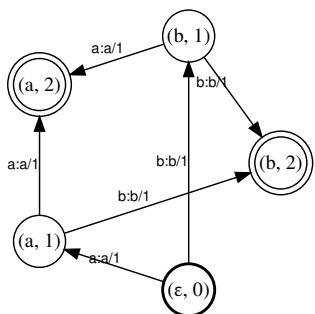

Figure 4: Illustration of the WFA $\mathcal{N}$ for $\Delta = \{a, b\}$, $p = 2$ and $l = 2$.

The initial state is $(\epsilon, 0)$ and the final states are those with the second element of the pair (the position) being $l$. Note that, by construction, $\mathcal{N}$ is deterministic. Then, the composition (or intersection) WFA $\mathcal{N} \circ \mathcal{M}$ still associates the same weight as $\mathcal{M}$ to each input string $y \in \Delta^l$. However, the states in that composition help us compute $\mathsf{L}(\mathbf{z}, s)$. In particular, for any $\mathbf{z} \in \Delta^p$ and $s \in [l]$, let $\mathsf{E}(\mathbf{z}, s)$ be the set of transitions of $\mathcal{N} \circ \mathcal{M}$ constructed by pairing the transition $((\mathbf{z}_1^{p-1}, s-1), z_p, \omega(\mathbf{z}, s), (\mathbf{z}_2^p, s))$ in $\mathcal{N}$ with a transition $(q_{\mathcal{M}}, z_p, \omega, q'_{\mathcal{M}})$ in $\mathcal{M}$. They admit the following form:

$$\mathsf{E}(\mathbf{z}, s) = \left\{ \left( (q_{\mathcal{N}}, q_{\mathcal{M}}), z_p, \omega, (q'_{\mathcal{N}}, q'_{\mathcal{M}}) \right) \in \mathsf{E}_{\mathcal{N} \circ \mathcal{M}} : q_{\mathcal{N}} = (\mathbf{z}_1^{p-1}, s-1) \right\}. \tag{35}$$

The WFA $\mathcal{N} \circ \mathcal{M}$ is deterministic as a composition of two deterministic WFAs. Thus, there is a unique path labeled with a sequence $y \in \Delta^l$ in $\mathcal{N} \circ \mathcal{M}$ and $y$ admits the substring $\mathbf{z}$ ending at position

$s$ iff that path goes through a transition in $\mathsf{E}(\mathbf{z}, s)$ when reaching position $s$. Therefore, to compute $\mathsf{L}(\mathbf{z}, s)$, it suffices for us to compute the sum of the weights of all paths in $\mathcal{N} \circ \mathcal{M}$ going through a transition in $\mathsf{E}(\mathbf{z}, s)$. This can be done straightforwardly using the forward-backward algorithm or two single-source shortest-distance algorithm over the $(+, \times)$ semiring [Mohri, 2002a], one from the initial state, the other one from the final states. Since $\mathcal{N} \circ \mathcal{M}$ is acyclic and admits $O(l|\Delta|^p)$ transitions, we can compute all the quantities $\mathsf{L}(\mathbf{z}, s)$, $s \in [l]$ and $\mathbf{z} \in \Delta^p$, in time $O(l|\Delta|^p)$.

**Markovian loss.** We consider adopting a Markovian assumption, which is commonly adopted in natural language processing [Manning and Schütze, 1999]. We will assume that $\bar{\ell}$ can be decomposed as follows for all $y, y' \in \Delta^l$: $\bar{\ell}(y, y') = \prod_{t=1}^{l} \bar{\ell}_t(y_{t-p+1}^t, y')$. Thus, we can write:

$$\mathsf{L}(\mathbf{z}, s) = \sum_{y: y_{s-p+1}^s = \mathbf{z}} \prod_{t=1}^{l} \bar{\ell}_t(y_{t-p+1}^t, y_i).$$

To efficiently compute $\mathsf{L}(\mathbf{z}, s)$, we will use a WFA representation similar to the one used by Cortes et al. [2016, 2018] and, for convenience, will adopt a similar notation. $\mathsf{L}(\mathbf{z}, s)$ coincides with a flow computation in a WFA $\mathcal{A}$ that we now define. $\mathcal{A}$ has the following set of states:

$$Q_{\mathcal{A}} = \left\{ (y_{t-p+1}^t, t) : y \in \Delta^l, t = 0, \ldots, l \right\},$$

with $\mathsf{I}_{\mathcal{A}} = (\varepsilon, 0)$ its single initial state, $\mathcal{F}_{\mathcal{A}} = \{(y_{l-p+1}^l, l) : y \in \Delta^l\}$ its set of final states, and a transition from state $(y_{t-p+1}^{t-1}, t-1)$ to state $(y_{t-p+2}^{t-1} b, t)$ with label $b$ and weight $\omega(y_{t-p+1}^{t-1} b, t) = \bar{\ell}_t(y_{t-p+1}^{t-1} b, y_i)$, that is the following set of transitions:

$$\mathsf{E}_{\mathcal{A}} = \left\{ \left( (y_{t-p+1}^{t-1}, t-1), b, \omega(y_{t-p+1}^{t-1} b, t), (y_{t-p+2}^{t-1} b, t) \right) : y \in \Delta^l, b \in \Delta, t \in [l] \right\}.$$

By construction, $\mathcal{A}$ is deterministic. The weight of a path in $\mathcal{A}$ is obtained by multiplying the weights of its constituent transitions. In view of that, $\mathsf{L}(\mathbf{z}, s)$ can be seen as the sum of the weights of all paths in $\mathcal{A}$ going through the transition from state $(\mathbf{z}_1^{p-1}, s-1)$ to $(\mathbf{z}_2^p, s)$ with label $z_p$.

For any state $(y_{t-p+1}^t, t) \in Q_{\mathcal{A}}$, we denote by $\alpha((y_{t-p+1}^t, t))$ the sum of the weights of all paths in $\mathcal{A}$ from the initial state $\mathsf{I}_{\mathcal{A}}$ to $(y_{t-p+1}^t, t)$ and by $\beta((y_{t-p+1}^t, t))$ the sum of the weights of all paths from $(y_{t-p+1}^t, t)$ to a final state. Then, $\mathsf{L}(\mathbf{z}, s)$ is given by

$$\mathsf{L}(\mathbf{z}, s) = \alpha\left((\mathbf{z}_1^{p-1}, s-1)\right) \times \omega(\mathbf{z}, s) \times \beta\left((\mathbf{z}_2^p, s)\right).$$

Since $\mathcal{A}$ is acyclic, $\alpha$ and $\beta$ can be computed for all states in linear time in the size of $\mathcal{A}$ using a single-source shortest-distance algorithm over the $(+, \times)$ semiring or the so-called forward-backward algorithm. Thus, since $\mathcal{A}$ admits $O(l|\Delta|^p)$ transitions, we can also compute all quantities $\mathsf{L}(\mathbf{z}, s)$, $s \in [l]$ and $\mathbf{z} \in \Delta^p$, in time $O(lr^p)$.

## G.2 Efficient gradient computation for $\mathsf{L}_{\exp}^{\text{comp}}$

In this section, we provide a brief overview of the gradient computation for $\mathsf{L}_{\exp}^{\text{comp}}$, which is similar to the approach used for $\mathsf{L}_{\log}^{\text{comp}}$.

Note that the loss $\mathsf{L}_{\exp}^{\text{comp}}$ on a given point $(x_i, y_i)$ can be expressed as follows:

$$
\begin{aligned}
\mathsf{L}_{\exp}^{\text{comp}} &= \sum_{y' \in \Delta^l} \bar{\ell}(y', y_i) \sum_{y'' \neq y'} e^{h(x_i, y'') - h(x_i, y')} \\
&= \sum_{y' \in \Delta^l} \bar{\ell}(y', y_i) \sum_{y'' \in \Delta^l} e^{h(x_i, y'') - h(x_i, y')} - \sum_{y' \in \Delta^l} \bar{\ell}(y', y_i) \\
&= \left[ \sum_{y'' \in \Delta^l} e^{h(x_i, y'')} \right] \sum_{y' \in \Delta^l} \bar{\ell}(y', y_i) e^{-h(x_i, y')} - \sum_{y' \in \Delta^l} \bar{\ell}(y', y_i) \\
&= \left[ \sum_{y'' \in \Delta^l} e^{\mathbf{w} \cdot \Psi(x_i, y'')} \right] \sum_{y' \in \Delta^l} \bar{\ell}(y', y_i) e^{-\mathbf{w} \cdot \Psi(x_i, y')} - \sum_{y' \in \Delta^l} \bar{\ell}(y', y_i).
\end{aligned}
$$

The gradient of $\mathsf{L}_{\exp}^{\text{comp}}$ is therefore given by

$$
\nabla\mathsf{L}_{\exp}^{\text{comp}}(\mathbf{w}) = \left[\sum_{y''\in\Delta^l} e^{\mathbf{w}\cdot\Psi(x_i,y'')}\Psi(x_i,y'')\right]\sum_{y'\in\Delta^l}\overline{\ell}(y',y_i)e^{-\mathbf{w}\cdot\Psi(x_i,y')}
$$
$$
-\left[\sum_{y''\in\Delta^l} e^{\mathbf{w}\cdot\Psi(x_i,y'')}\right]\sum_{y'\in\Delta^l}\overline{\ell}(y',y_i)e^{-\mathbf{w}\cdot\Psi(x_i,y')}\Psi(x_i,y').
$$

(36)

An efficient computation of these terms is not straightforward since the summations run over an exponential number of sequences for $y$. However, we will leverage the Markovian property of the features to design an efficient computation. This approach is similar to what we demonstrated earlier for $\mathsf{L}_{\log}^{\text{comp}}$. We start with identifying the most computationally challenging terms by rewriting the expression of the gradient of $\mathsf{L}_{\exp}^{\text{comp}}$ in the following lemma.

**Lemma 19.** *For any $\mathbf{w}\in\mathbb{R}^d$, the gradient of $\mathsf{L}_{\exp}^{\text{comp}}$ can be expressed as follows:*

$$
\nabla\mathsf{L}_{\exp}^{\text{comp}}(\mathbf{w}) = \sum_{s=1}^{l}\sum_{\mathbf{z}\in\Delta^p}[\mathsf{N}_{\mathbf{w}}\mathsf{Q}'_{\mathbf{w}}(\mathbf{z},s) - Z_{\mathbf{w}}\mathsf{C}_{\mathbf{w}}(\mathbf{z},s)]\widetilde{\psi}(x_i,\mathbf{z},s),
$$

*where $\mathsf{Q}'_{\mathbf{w}}(\mathbf{z},s) = \sum_{y:y_{s-p+1}^s=\mathbf{z}} e^{\mathbf{w}\cdot\Psi(x_i,y)}$, $\mathsf{C}_{\mathbf{w}}(\mathbf{z},s) = \sum_{y:y_{s-p+1}^s=\mathbf{z}}\overline{\ell}(y,y_i)e^{-\mathbf{w}\cdot\Psi(x_i,y)}$ and $\mathsf{N}_{\mathbf{w}} = \sum_{y\in\Delta^l}\overline{\ell}(y,y_i)e^{-\mathbf{w}\cdot\Psi(x_i,y)}$.*

*Proof.* Using the decomposition of the feature vector, we can write:

$$
\sum_{y\in\Delta^l} e^{\mathbf{w}\cdot\Psi(x_i,y)}\Psi(x_i,y) = \sum_{y\in\Delta^l} e^{\mathbf{w}\cdot\Psi(x_i,y)}\sum_{s=1}^{l}\widetilde{\psi}(x_i,y_{s-p+1}^s,s)
$$
$$
= \sum_{s=1}^{l}\sum_{\mathbf{z}\in\Delta^p}\left[\sum_{y:y_{s-p+1}^s=\mathbf{z}} e^{\mathbf{w}\cdot\Psi(x_i,y)}\right]\widetilde{\psi}(x_i,\mathbf{z},s)
$$
$$
\sum_{y\in\Delta^l}\overline{\ell}(y,y_i)e^{-\mathbf{w}\cdot\Psi(x_i,y)}\Psi(x_i,y) = \sum_{y\in\Delta^l}\overline{\ell}(y,y_i)e^{-\mathbf{w}\cdot\Psi(x_i,y)}\sum_{s=1}^{l}\widetilde{\psi}(x_i,y_{s-p+1}^s,s)
$$
$$
= \sum_{s=1}^{l}\sum_{\mathbf{z}\in\Delta^p}\left[\sum_{y:y_{s-p+1}^s=\mathbf{z}}\overline{\ell}(y,y_i)e^{-\mathbf{w}\cdot\Psi(x_i,y)}\right]\widetilde{\psi}(x_i,\mathbf{z},s).
$$

This completes the proof. $\qquad\square$

It was shown by Cortes et al. [2016, 2018] that all of the quantities $\mathsf{Q}'_{\mathbf{w}}(\mathbf{z},s)$ for $\mathbf{z}\in\Delta^p$ and $s\in[l]$ and $Z_{\mathbf{w}}$ can be computed efficiently in time $O(lr^p)$, where $r=|\Delta|$. Thus, the remaining bottleneck in the gradient computation suggested by Lemma 19 is the evaluation of the quantities $\mathsf{C}_{\mathbf{w}}(\mathbf{z},s)$ for $\mathbf{z}\in\Delta^p$ and $s\in[l]$ and $\mathsf{N}_{\mathbf{w}}$. As with the loss function $\mathsf{L}_{\log}^{\text{comp}}$ discussed in the previous section, we will analyze the computation of these quantities first in the case of rational losses, next in that of Markovian loss.

**Rational losses.** We will adopt the same notation as in the case of the $\mathsf{L}_{\log}^{\text{comp}}$ loss with the same definition of a *rational loss*: $\ell$ is a rational loss if there exists a WFST over the $(+,\times)$ semiring over the reals with $\Delta$ as both the input and output alphabet such that for all $y,y'\in\Delta^*$, we have $\overline{\ell}(y,y') = \mathcal{U}(y,y')$.

Our algorithm is also somewhat similar to the one described for the $\mathsf{L}_{\log}^{\text{comp}}$ loss or that of Cortes et al. [2018] for the computation of the gradient of the VCRF loss function. There are, however, several differences here too because the quantities computed and thus the automata operations required are distinct.

Exactly as in the case of $\mathsf{L}_{\log}^{\text{comp}}$ loss, we first define a deterministic WFA $\mathcal{M}$ over the $(+,\times)$ semiring that can be computed in polynomial time and that admits a unique path labeled with any sequence

$y \in \Delta^l$, whose weight is $\mathcal{U}(y, y_i)$. Next, as in [Cortes et al., 2018], we define a deterministic WFA $\mathcal{A}$ such that

$$\mathcal{A}(y) = e^{-\mathbf{w} \cdot \Psi(x_i, y)} = \prod_{s=1}^{l} e^{-\mathbf{w} \cdot \widetilde{\psi}(x_i, y_{s-p+1}^s, s)}.$$

The set of states $Q_{\mathcal{A}}$ of $\mathcal{A}$ are defined as $Q_{\mathcal{A}} = \left\{ (y_{s-p+1}^s, s) : y \in \Delta^l, s = 0, \ldots, l \right\}$, with $I_{\mathcal{A}} = (\varepsilon, 0)$ its single initial state, $\mathcal{F}_{\mathcal{A}} = \{ (y_{l-p+1}^l, l) : y \in \Delta^l \}$ its set of final states, and with a transition from state $(y_{s-p+1}^{s-1}, s-1)$ to state $(y_{s-p+2}^{s-1} a, s)$ with label $a$ and weight, that is, the following set of transitions:

$$\mathsf{E}_{\mathcal{A}} = \left\{ \left( (y_{s-p+1}^{s-1}, s-1), a, e^{-\mathbf{w} \cdot \widetilde{\psi}(x_i, y_{s-p+1}^{s-1} a, s)}, (y_{s-p+2}^{s-1} a, s) \right) : y \in \Delta^l, a \in \Delta, s \in [l] \right\}.$$

Then, by definition of composition or intersection [Mohri, 2009], the WFA $(\mathcal{M} \circ \mathcal{A})$ is deterministic and admits a unique path labeled with any given $y \in \Delta^l$ whose weight is $(\mathcal{M} \circ \mathcal{A})(y) = \mathcal{M}(y) \cdot \mathcal{A}(y) = \overline{\ell}(y, y_i) e^{-\mathbf{w} \cdot \Psi(x_i, y)}$.

Now, $\mathsf{N_w}$ coincides with the sum of the weights of all accepted paths in this WFA. Thus, since $(\mathcal{M} \circ \mathcal{A})$ is acyclic, it can be computed in time linear in the size of $(\mathcal{M} \circ \mathcal{A})$, that is its number of transitions. For any $s \in [l]$ and $\mathbf{z} \in \Delta^p$, $\mathsf{C_w}(\mathbf{z}, s)$ is the sum of the weights of all paths in $(\mathcal{M} \circ \mathcal{A})$ labeled with a sequence $y$ admitting $\mathbf{z}$ as a substring ending at position $s$. The states in the composition $(\mathcal{M} \circ \mathcal{A})$ help us compute

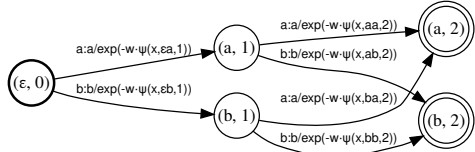

Figure 5: Illustration of the WFA $\mathcal{A}$ for $\Delta = \{a, b\}$, $p = 2$ and $l = 2$.

$\mathsf{C_w}(\mathbf{z}, s)$. As in the case of the $\mathsf{L}_{\log}^{\text{comp}}$ loss, for any $\mathbf{z} \in \Delta^p$ and $s \in [l]$, we define $\mathsf{E}(\mathbf{z}, s)$ as the set of transitions of $(\mathcal{M} \circ \mathcal{A})$ constructed by pairing a transition $(q_{\mathcal{M}}, z_p, \omega_{\mathcal{M}}, q'_{\mathcal{M}})$ in $\mathcal{M}$ with the transition $((\mathbf{z}_1^{p-1}, s-1), z_p, \omega(\mathbf{z}, s), (\mathbf{z}_2^p, s))$ in $\mathcal{A}$. They admit the following form:

$$\mathsf{E}(\mathbf{z}, s) = \left\{ \left( (q_{\mathcal{M}}, q_{\mathcal{A}}), z_p, \omega_{\mathcal{M}} \cdot \omega(\mathbf{z}, s), (q'_{\mathcal{M}}, q'_{\mathcal{A}}) \right) \in \mathsf{E}_{\mathcal{M} \circ \mathcal{A}} : q_{\mathcal{A}} = (\mathbf{z}_1^{p-1}, s-1) \right\}. \tag{37}$$

The WFA $(\mathcal{M} \circ \mathcal{A})$ is deterministic as a composition of two deterministic WFAs. Thus, there is a unique path labeled with a sequence $y \in \Delta^l$ in $(\mathcal{M} \circ \mathcal{A})$ and $y$ admits the substring $\mathbf{z}$ ending at position $s$ iff that path goes through a transition in $\mathsf{E}(\mathbf{z}, s)$ when reaching position $s$. Therefore, to compute $\mathsf{C_w}(\mathbf{z}, s)$, it suffices for us to compute the sum of the weights of all paths in $\mathsf{C_w}(\mathbf{z}, s)$ going through a transition in $\mathsf{E}(\mathbf{z}, s)$. This can be done straightforwardly using the forward-backward algorithm or two single-source shortest-distance algorithm over the $(+, \times)$ semiring [Mohri, 2002a], one from the initial state, the other one from the final states. Since $(\mathcal{M} \circ \mathcal{A})$ is acyclic and admits $O(l|\Delta|^p)$ transitions, we can compute all the quantities $\mathsf{C_w}(\mathbf{z}, s)$, $s \in [l]$ and $\mathbf{z} \in \Delta^p$, in time $O(l|\Delta|^p)$.

**Markovian loss.** Here, we adopt the Markovian assumption and assume that $\overline{\ell}$ can be decomposed as follows for all $y, y' \in \Delta^l$: $\overline{\ell}(y, y') = \prod_{t=1}^{l} \overline{\ell}_t(y_{t-p+1}^t, y')$. Thus, the quantity $\mathsf{C_w}(\mathbf{z}, s)$ can be written as:

$$\mathsf{C_w}(\mathbf{z}, s) = \sum_{y : y_{s-p+1}^s = \mathbf{z}} \prod_{t=1}^{l} \overline{\ell}_t(y_{t-p+1}^t, y_i) e^{-\mathbf{w} \cdot \sum_{k=1}^{l} \widetilde{\psi}(x_i, y_{k-p+1}^k, k)}$$

$$= \sum_{y : y_{s-p+1}^s = \mathbf{z}} \prod_{t=1}^{l} \overline{\ell}_t(y_{t-p+1}^t, y_i) \prod_{k=1}^{l} e^{-\mathbf{w} \cdot \widetilde{\psi}(x_i, y_{k-p+1}^k, k)}$$

$$= \sum_{y : y_{s-p+1}^s = \mathbf{z}} \prod_{t=1}^{l} \overline{\ell}_t(y_{t-p+1}^t, y_i) e^{-\mathbf{w} \cdot \widetilde{\psi}(x_i, y_{t-p+1}^t, t)}.$$

Then, we can proceed as in the Markovian loss case for the loss function $\mathsf{L}_{\log}^{\text{comp}}$ except that instead of the WFA $\mathcal{A}$ used there, we define here a similar WFA $\mathcal{A}'$. The only difference is that the weight $\omega(y_{t-p+1}^{t-1} b, t) = \overline{\ell}_t(y_{t-p+1}^{t-1} b, y_i)$ for the WFA $\mathcal{A}$ is replaced with $\omega'(y_{t-p+1}^{t-1} b, t) = \overline{\ell}_t(y_{t-p+1}^{t-1} b, y_i) e^{-\mathbf{w} \cdot \widetilde{\psi}(x_i, y_{t-p+1}^{t-1} b, t)}$. With the same argument, we can compute all quantities $\mathsf{C_w}(\mathbf{z}, s)$, $s \in [l]$ and $\mathbf{z} \in \Delta^p$, in time $O(l r^p)$. The quantity $\mathsf{N_w}$ can also be efficiently computed in time $O(l r^p)$ since it is the sum of the weights of all paths in $\mathcal{A}'$.

### G.3 Efficient Inference

We focused on the problem of efficient computation of the gradient. Inference is also a key problem in structured prediction since the label with a highest score must be determined out of an exponentially large set of possible ones. However, for the linear hypotheses considered in the previous sections, this problem can be efficiently tackled since it can be cast as a shortest-distance problem in a directed acyclic graph, as in [Cortes, Kuznetsov, Mohri, and Yang, 2016].

More generally, an efficient gradient computation, efficient inference and other related algorithms can benefit from standard weighted automata and transducer optimization algorithms such as $\epsilon$-removal [Mohri, 2000, 2002b] and determinization [Mohri, 1997, Mohri and Riley, 1997, Allauzen and Mohri, 2003, 2004] (see also the survey chapter [Mohri, 2009]).

