# OpenReview forum: "Structured Prediction with Stronger Consistency Guarantees"
_NeurIPS.cc/2023/Conference — NeurIPS 2023 poster_

### Official Review · Reviewer_oyCT · 2023-06-24

**Soundness:** 3 good
**Presentation:** 3 good
**Contribution:** 3 good
**Rating:** 7
**Confidence:** 4

**Summary:**

This work studies $\mathcal{H}$-consistency of surrogate losses for structured prediction. The authors show that non classic surrogate losses are not $\mathcal{H}$-consistent thus not Bayes-consistent. They propose two families of $\mathcal{H}$-consistent losses as extensions to existing losses and algorithms for two special cases of loss.



**Strengths:**

The manuscript is well-written with few typos. The theoretical results are sound and novel. Practical optimization methods are provided to show that the proposed losses are not only consistent but also computationally feasible.

**Weaknesses:**

I don't see obvious weaknesses in the manuscript. A possible drawback might be the lack of empirical results.

**Questions:**

I have a few minor concerns as follows.

1. The inconsistency results for each individual classic loss in Section 3 are not new but the authors seem to give a novel unified view of four losses. Does this include Fenchel-Young losses?

2. The consistency bounds in Theorem 6 and 8 depend on a quantity defined by the minimizability gap, which is difficult or impossible to minimize or estimate. I understand that a non-asymptotic bound is desirable but isn't it implicit since we may not obtain an estimated upper bound for a special case unlike what we can do with a generalization bound based on Rademacher complexities?

3. Can you discuss and compare $\mathcal{H}$-consistency with Fisher consistency since there is a requirement for Fisher consistency in terms of a comparison inequality? (Nowak et al., 2020) (Blondel, 2019)

4. Section E presents proofs for each of the four structured comp-sum losses individually but Theorem 6 is stated for general structured comp-sum losses. Does the conclusion hold or did I miss something?

5. Line 270, $\bar{\ell}_i$ instead of $\ell_i$.

6. I am confused with the definitions about Markovian features in lines 287-294. Specifically, what's the definition of $p$? Based on the context, I can infer that $\mathbf{\Psi}$, $\mathbf{\psi}$, $\mathbf{\Psi}_k$, $\mathbf{\psi}_k$, $\tilde{\psi}$ all refer to a vector in $\mathbb{R}^d$. But what is the number of padding zeros in the definition of $\tilde{\mathbf{\Psi}}_k$ in line 290?

7. Due to the theoretical nature of this work, experiments may not be necessary. But is it possible to compare the proposed consistent losses with other consistent losses empirically (Nowak et al., 2020) (Blondel, 2019)?


**Limitations:**

Limitations are mentioned together with introduction of methods. Potential negative societal impacts are not discussed.

---

> ### Author Rebuttal · Authors · 2023-08-08
>
> Thank you for your appreciation of our work. We will take your suggestions into account when preparing the final version. Below please find responses to specific questions.
>
> **Questions:**
>
> **1. The inconsistency results for each individual classic loss in Section 3 are not new but the authors seem to give a novel unified view of four losses. Does this include Fenchel-Young losses?**
>
> **Response:** The formulation presented in Section 3 differs from a general formulation of Fenchel-Young losses (or Bregman divergence losses), even though they both encompass specific instances. For example, both include the hinge loss of support vector machines.
>
> **2. The consistency bounds in Theorem 6 and 8 depend on a quantity defined by the minimizability gap, which is difficult or impossible to minimize or estimate. I understand that a non-asymptotic bound is desirable but isn't it implicit since we may not obtain an estimated upper bound for a special case unlike what we can do with a generalization bound based on Rademacher complexities?**
>
> **Response:** That’s a natural question. Let us first mention that the minimizability gap can be crudely upper bounded by the approximation error. But, more significantly, although we have not detailed it in this paper, the minimizability gap can in fact be upper bounded in terms of useful terms depending on the magnitude of the parameter space. Our $H$-consistency bounds can be used to derive finite sample learning bounds for a hypothesis set $H$ expressed in terms of the Rademacher complexity of the hypothesis set and the loss function and an upper bound on the minimizability gap for the surrogate loss. We will elaborate on that in the final version.
>
> **3. Can you discuss and compare $H$-consistency with Fisher consistency since there is a requirement for Fisher consistency in terms of a comparison inequality? (Nowak et al., 2020) (Blondel, 2019)**
>
> **Response:** With Nowak et al.'s (2020) definition, Fisher consistency coincides with the specific case of $H$-consistency, where $H$ is the family of all measurable functions. However, when dealing with a constrained hypothesis set $H$, the comparison inequality does not yield an $H$-consistency bound that relates the surrogate estimation loss to the target estimation loss in terms of minimizability gaps.
>
> **4. Section E presents proofs for each of the four structured comp-sum losses individually but Theorem 6 is stated for general structured comp-sum losses. Does the conclusion hold or did I miss something?**
>
> **Response:** Theorem 6 represents a consolidated result for the four structured comp-sum losses, with the proofs for each being presented separately in Section E. We will clarify this distinction in the final version.
>
> **5. Line 270, $\bar \ell_i$ instead of $\ell_i$.**
>
> **Response:** Thanks, we will correct that.
>
> **6. I am confused with the definitions about Markovian features in lines 287-294. Specifically, what's the definition of $p$? Based on the context, I can infer that $\Psi$, $\psi$, $\Psi_k$, $\psi_k$, $\tilde{\psi}$ all refer to a vector in $\mathbb{R}^d$. But what is the number of padding zeros in the definition of $\tilde{\Psi}_k$ in line 290?**
>
> **Response:** Each $\Psi_k$ corresponds to a Markovian feature vector based only on $k$-grams, $p$ is the largest $k$. We will fully clarify the notation in the final version.
>
> **7. Weaknesses: I don't see obvious weaknesses in the manuscript. A possible drawback might be the lack of empirical results.**
>
> **Due to the theoretical nature of this work, experiments may not be necessary. But is it possible to compare the proposed consistent losses with other consistent losses empirically (Nowak et al., 2020) (Blondel, 2019)?**
>
> **Response:** Thank you for the suggestion. We will seek to add such experiments in the final version on the empirical comparison with other consistency losses in previous work. But, as you have mentioned, our paper mainly focuses on the theoretically principled surrogate losses for structured prediction based on $H$-consistency bounds. While we have demonstrated that the minimization of several proposed loss functions such as structured logistic loss benefit from efficient algorithms, we recognize the importance of further exploration. As such, we intend to dedicate future work to an extensive empirical analysis and the development of more universally applicable algorithmic solutions to encompass a broader family of surrogate loss functions as well as a diverse range of target losses.

---

> > ### Comment · Reviewer_oyCT · 2023-08-20
> >
> > Thanks for your response, which has addressed all my concerns. Please make sure Question 2 and 4 are addressed in your revision which I believe should be helpful to readers. I will maintain my score and vote for acceptance.

---

### Official Review · Reviewer_y5Sa · 2023-07-02

**Soundness:** 3 good
**Presentation:** 3 good
**Contribution:** 3 good
**Rating:** 7
**Confidence:** 3

**Summary:**

This work extensively studies surrogate losses for structured predictions supported by H-consistency bounds. It first shows several negative results for some widely used surrogate losses in structured predictions: no non-trivial H-consistency bound can be derived. Then it provides two new families of surrogate losses that are supported by H-consistency bounds (which imply Bayes consistency): structured comp-sum losses and structured constrained losses. Finally, efficient algorithms are proposed for some of these new surrogate losses.

**Strengths:**

This work presents a solid theoretical study in the field of structured predictions. First, the paper shows that structured max losses (which include loss functions associated with several prominent structured prediction algorithms in the literature) are not Bayes consistent, which implies they cannot be supported by H-consistency bounds either (Theorem 4). Then, it shows that voted Conditional Random Field losses (which have been presented in some works of structured predictions) are not Bayes consistent either (Theorem 5). Moving to positive results, the paper provides two new families of surrogate losses that are supported by H-consistency bounds (which imply Bayes consistency): structured comp-sum losses and structured constrained losses (Theorem 6, 8, Corollary 7, 9). Finally, it presents efficient algorithms for minimizing several of the proposed surrogate losses.

The level of originality exhibited in the research was noticeable, demonstrating a study (H-consistency bounds in structured predictions) that has not been extensively explored before. The quality of the work is good, with meticulous comparisons and contrasts with previous works (including some prominent ones) in structured predictions (Line 134-145, 157-162) and detailed analysis, supporting the authors' arguments convincingly. The paper is well-written at large (some suggestions for improvements below). The work is significant; it offers valuable insights into an under-researched topic (H-consistency bounds in structured predictions), and the implications of the findings could stimulate new directions for future research in structured predictions.

**Weaknesses:**

1. The current manuscript does not have a conclusion section. Given that the results are already impressive, the authors should have shortened Section 6 and added a conclusion section to improve the readability further.
2. Because the authors showed the gradient of the structured logistic loss can be computed efficiently and claimed practical use, the authors should consider including some experiments for demonstration.

**Questions:**

1. When you say "no non-trivial H-consistency bounds can be derived", what are the "trivial H-consistency bounds"?
2. How are the results of this work related to those in several works by Ciliberto et al., 2016, 2019, and 2020?

**Limitations:**

I could not find the location where the limitations of the work were explicitly discussed.

---

> ### Author Rebuttal · Authors · 2023-08-08
>
> Thank you for your appreciation of our work. We will take your suggestions into account when preparing the final version. Below please find responses to specific questions.
>
> **Weaknesses:**
>
> **1. The current manuscript does not have a conclusion section. Given that the results are already impressive, the authors should have shortened Section 6 and added a conclusion section to improve the readability further.**
>
> **Response:** Thank you for the suggestion, we will definitely add a conclusion section.
>
> **2. Because the authors showed the gradient of the structured logistic loss can be computed efficiently and claimed practical use, the authors should consider including some experiments for demonstration.**
>
> **Response:** We will take into account your suggestions when preparing the final version. Our paper mainly focuses on the theoretically principled surrogate losses for structured prediction based on $H$-consistency bounds. While we have demonstrated that the minimization of several proposed loss functions such as structured logistic loss benefit from efficient algorithms, we recognize the importance of further exploration. As such, we intend to dedicate future work to an extensive empirical analysis and the development of more universally applicable algorithmic solutions to encompass a broader family of surrogate loss functions as well as a diverse range of target losses.
>
> **Questions:**
>
> **1. When you say "no non-trivial H-consistency bounds can be derived", what are the "trivial H-consistency bounds"?**
>
> **Response:** We refer to a bound as in (4) where $f(t)$ does not tend to zero as $t$ approaches zero, for example because it is lower bounded by a constant. In such cases, the bound becomes uninformative about the left-hand side even when the argument of $f$ is small, and it does not even guarantee Bayes-consistency when $H$ represents the family of all measurable functions. We will further clarify on this matter in the final version.
>
> **2. How are the results of this work related to those in several works by Ciliberto et al., 2016, 2019, and 2020?**
>
> **Response:** Ciliberto et al. [2016] focused on a least squares surrogate loss function and corresponding framework. In this framework, the structured prediction problem is cast as a regression problem. They derived a regularization approach to structured prediction from the least squares surrogate loss and proved the Bayes-consistency of that approach. Ciliberto et al. [2019] focused on a local structure-adapted framework for structured prediction. They proposed a novel structured prediction algorithm that adaptively leverages locality in the learning problem. Ciliberto et al. [2020] developed a general framework for structured prediction based on implicit embedding. Their methods lead to polyhedral-type surrogates losses that benefit from Bayes-consistency.
>
> On the other hand, our work presents an extensive study of surrogate losses for structured prediction supported by $H$-consistency bounds. Different from the surrogate losses studied in the previous work, the formulations of our proposed surrogate losses including structured comp-sum losses and structured constrained losses are completely novel and do not cast structured prediction problems as a regression problem. Furthermore, we prove stronger consistency guarantees that imply Bayes-consistency for these new proposed families of surrogate loss. We will further clarify and detail these comparisons in the final version.
>
> **Limitations:**
> **I could not find the location where the limitations of the work were explicitly discussed.**
>
> **Response:** Thank you for pointing it out. We will add a separate discussion on potential limitations in the final version.

---

> > ### Comment · Reviewer_y5Sa · 2023-08-16
> >
> > Thank the authors for their responses. I have also read other reviews. I stand by my initial rating.

---

### Official Review · Reviewer_wtd4 · 2023-07-06

**Soundness:** 3 good
**Presentation:** 3 good
**Contribution:** 3 good
**Rating:** 6
**Confidence:** 3

**Summary:**

In this paper, the authors study surrogate losses for structured prediction problems. They show that surrogate losses proposed in previous work are not Bayes-consistent, i.e. a sequence of hypotheses which minimises the surrogate loss may not minimise the target loss. They then introduce two families of surrogate losses, namely structured comp-sum losses and structured constrained loss functions, which generalise two corresponding existing families to the structured prediction setting, and show that these admit $\\mathcal{H}$-consistency bounds, a stronger form of consistency that also implies Bayes-consistency. Last, they derive practically efficient algorithms for minimising the surrogate losses they have introduced, under certain settings.

**Strengths:**

__Paper highlights shortcomings of existing theory:__ The authors prove that existing surrogate loss functions, namely the structured max loss and the structured voted conditional random field loss, are not Bayes-consistent. Therefore, minimising one of these surrogates is not guaranteed to also minimise the corresponding target losses. This result is likely of interest to the wider community.

__Introduction of new structured losses supported by theory:__ This work introduces two classes of surrogate losses, namely the structured comp-sum and the structured constrained loss, which they show to be $\\mathcal{H}$-consistent. This implies Bayes consistency, a guarantee that was missing from existing structured surrogate losses. In fact, $\\mathcal{H}$-consistency is a stronger guarantee than Bayes-consistency, since, as the authors point out, it is not asymptotic and accounts for the hypothesis set $\\mathcal{H}$ in question.

__Theoretically sound and technically precise:__ Although I was not able to check the entirety of the derivations in the Appendix (I did not verify the proofs of Theorem 6 and Theorem 8 closely enough to be fully confident about these), other parts which I did check closely looked sound and technically precise to me. Also, generally, I found the paper to be carefully written and I also found that the notation was precise and clear (though I thought Section 6 could have been better organised to improve readability, and that the paper could have benefited from a conclusion section).

**Weaknesses:**

__The exposition of the paper could be improved:__ While I found that the notation and definitions in the paper are clear and precise, I found that from Section 6 onwards, the paper has tougher to follow, and the exposition was far denser and less clear than in the preceding sections. I think that the paper could benefit by organising the key results in lemmas and propositions, and deferring more some of the details from this section into the appendix.


**Questions:**

Please see a list of questions and suggestions for improvement I have about the paper, organised roughly in order of appearance:

- __Line 83:__ The authors say “naturally $\\ell$ is symmetric”. Does it have to be symmetric, or are they assuming it to be symmetric?

- __Lines 95-97:__ The phrasing here could perhaps be clarified to “which guarantees that minimizing the generalization error for a surrogate loss $L_{\\text{sur}}$ over $\\mathcal{H}_{\\text{all}}$ also leads to the minimization of generalization error for the target loss $L$.”

- __Definition 1:__ This appears to have typos. First $f_n$ appears in eq. 3 but it is not defined. Second, $h$ appears in the definition but not in eq. 3. Do the authors mean $h_n$ in the definition and $h_n$ instead of $f_n$ in eq. 3?

- __Definition 2:__ How is $\\mathcal{H}$ defined (in line 107)? Is this a subset of the full hypothesis set? If so, the authors could clarify this by saying “Given a subset of the hypothesis class $\\mathcal{H} \\subseteq \\mathcal{H}_{\\text{all}}$, a surrogate loss…”.

- __Equation below line 116:__ Should the expression on the right hand side have the conditional distribution $p(y | x)$ rather than the joint $p(x, y)$?

- __Comment on notation:__ In line 117 the authors could use the notation $\\mathcal{C}_{L, \\mathcal{H}}^*(x)$ to make this consistent with the notation under eq. 79.

- __Line 122:__ Typo, “hypothesis set” not “hypothesis sets”.

- __Line 129:__ Typo, the “possible predictions” instead of “possible prediction”.

- __Lemma 3:__ Similarly to the above comment for line 116, should the lemma involve the conditional $p(y | x)$ rather than the joint $p(x, y)$? Furthermore, the authors don’t seem to be using Lemma 3 (or referring to it) in the rest of the main text. In this case, I would suggest removing it and deferring it to the appendix.

- __Theorem 4:__ I have two questions regarding Theorem 4:
    - First, in line 650, Appendix C, you refer to $h^*$ as “the Bayes classifier” of the structured max loss. I think a more accurate statement would be that $h^*$ is “__a__ Bayes classifier”, because there exist many choices of $h$ which minimise the generalisation error. For example, consider $h^*(x, 1) = h^*(x, 2) = 1$ and $h^*(x, y) = 0$ for all other $y > 2$. This also minimises the generalisation error, and coincides with the Bayes classifier of the target loss.

    - Second, the proof of this result seems to highlight a way in which the problematic classifiers (i.e. the classifiers which are optimal for the surrogate loss, but not for the target loss) are far fewer than those which are optimal for the target loss. In particular, is it correct to say that the set of classifiers which optimise the surrogate loss but not the target loss are those for which $h(x, y_1) = h(x, y_2) = \\dots = h(x, y_n)$, and this set is much smaller than the set of classifiers which optimise both the surrogate as well as the target loss (that is the set of classifiers for which $h(x, y_i) > h(x, y_3), \dots, h(x, y_n)$ for either $i = 1$ or $i = 2$). Can the authors comment on this point? For example, could one learn $h$ with an algorithm that involves a randomisation step (e.g. randomised initialisation), that results in convergence to the Bayes-optimal predictor of the target loss with high probability?

- __Line 663:__ In Appendix D, in the proof for Theorem 5, the authors introduce the quantity $\\Phi_y$. Is this simply a function from $\\mathcal{Y}$ to $\\mathbb{R}$?

- __Theorem 5:__ The proof of Theorem 5 relies on an argument (line 663) where the authors consider a loss $\ell(y’, y)$ that decouples (i.e. factorises) into a term that depends solely on $y$ only and another term which depends solely on $y’$. From this, they show that the VCRF loss is not Bayes-consistent. Some common losses, such as the zero-one $\\ell_{0-1}$ loss do not decompose in this way. When constrained to such losses, is the VCRF loss Bayes-consistent? How critical is the factorisation requirement in the proof?

**Limitations:**

In my assessment, I do not see any substantial limitations of this work which have not been addressed by the authors. However, I would appreciate the authors’ clarification on the questions I have raised above regarding the argument in the proofs of Theorems 4 and 5.

---

> ### Author Rebuttal · Authors · 2023-08-08
>
> Thank you for your thoughtful feedback and suggestions on improving the readability. We will take them all into account when preparing the final version. Below please find responses to specific questions.
>
> **Weaknesses:**
> **The exposition of the paper could be improved: While I found that the notation and definitions in the paper are clear and precise, I found that from Section 6 onwards, the paper has tougher to follow, and the exposition was far denser and less clear than in the preceding sections. I think that the paper could benefit by organising the key results in lemmas and propositions, and deferring more some of the details from this section into the appendix.**
>
> **Response:** Thank you for your suggestions.  We recognize that Section 6 may seem dense due to the necessity of introducing new notations and technical tools for efficient gradient computation and inference in structured prediction. We will follow your suggestion regarding the organization and will simplify our presentation to make it more accessible to readers. The addition of an extra page in the final version will also enable us to include more discussions in the main body, further enhancing the clarity and depth of our work.
>
> **Questions:**
>
> **1. Line 83: The authors say “naturally is symmetric”. Does it have to be symmetric, or are they assuming it to be symmetric?**
>
> **Response:** Yes, $\ell$ is assumed to be symmetric in our analysis. We will clarify that. We meant that this is a natural assumption since all instances of $\ell$ that we are familiar with in structured prediction admit this property.
>
> **2. Lines 95-97: The phrasing here could perhaps be clarified to “which guarantees that minimizing the generalization error for a surrogate loss $L_{\mathrm{sur}}$ over $H_{\mathrm{all}}$ also leads to the minimization of generalization error for the target loss $L$.”**
>
> **Response:** Thank you for the suggestion. We will clarify that in the final version.
>
> **3. Definition 1: This appears to have typos. First $f_n$ appears in eq. 3 but it is not defined. Second, $h$ appears in the definition but not in eq. 3. Do the authors mean $h_n$ in the definition and $h_n$ instead of $f_n$ in eq. 3?**
>
> **Response:** Thank you for pointing that out. You are indeed correct: $h$ and $f_n$ should be corrected to be $h_n$ in the definition. We will fix that in the final version.
>
> **4. Definition 2: How is $H$ defined (in line 107)? Is this a subset of the full hypothesis set? If so, the authors could clarify this by saying …**
>
> **Response:** Yes, $H$ is a subset of the family of all measurable functions. We will clarify that following your suggestion.
>
> **5. Equation below line 116: Should the expression on the right hand side have the conditional distribution $p(y | x)$ rather than the joint $p(x,y)$?**
>
> **Response:** Sorry for the confusion. We use the notation $p(x, y)$ to denote the conditional distribution as mentioned in line 115. We will make it more clear in the final version.
>
> **6. Comment on notation: In line 117 the authors could use the notation $\mathcal{C}^{*}_{L,H}(x)$ to make this consistent with the notation under eq. 79.**
>
> **Response:** Thanks, we will take your suggestions into account.
>
> **7. Line 122: Typo, “hypothesis set” not “hypothesis sets”.**
>
> **8. Line 129: Typo, the “possible predictions” instead of “possible prediction”.**
>
> **Response:** Thank you, we will correct these typos.
>
> **9. Lemma 3: Similarly to the above comment for line 116, should the lemma involve the conditional $p(y | x)$ rather than the joint $p(x,y)$? Furthermore, the authors don’t seem to be using Lemma 3 (or referring to it) in the rest of the main text. In this case, I would suggest removing it and deferring it to the appendix.**
>
> **Response:** Sorry for the confusion. We use the notation $p(x, y)$ to denote the conditional distribution as mentioned in line 115. We will make it more clear in the final version. We will consider moving Lemma 3 into the appendix following your suggestion.
>
> **10. Two questions regarding Theorem 4.**
>
> **Response:** With regard to your first question, you are indeed correct; the use of "a Bayes classifier" is more fitting in this context.
>
> As for your second question, it is definitely an intriguing one! You are right that in the given example, the set of classifiers that optimize the surrogate loss without optimizing the target loss is much smaller than the set that optimizes both. However, the applicability of this observation to general problems remains unclear. The idea of randomization in this context is indeed natural and potentially fruitful. We have explored a similar randomization idea in a different context without success but it is certainly a valuable avenue for further research.
>
>
> **11. Line 663: In Appendix D, in the proof for Theorem 5, the authors introduce the quantity $\Phi_y$. Is this simply a function from $\mathcal{Y}$ to $\mathbb{R}$?**
>
> **Response:** Yes, that’s right. We will further clarify that.
>
> **12. The proof of Theorem 5.**
>
> **Response:** That's a great question. In Theorem 5, we examine Bayes consistency within the context of structured prediction. This refers to the consistency property that must be maintained across any target loss function, as described in Definition 1. In our current proofs, we use the decoupling property of the target loss as a convenient technical assumption, facilitating the demonstration that surrogate loss and target loss lead to different Bayes classifiers. We believe that our proof can be extended beyond this assumption to encompass a more extensive family of target loss functions. However, it is definitely intriguing to investigate the consistency question further, especially when restricted to specific target loss functions.

---

> ### Comment · Reviewer_wtd4 · 2023-08-18
> **Thank you for your response**
>
> I would like to thank the authors for their response to my rebuttal. I have read through this and was pleased to see that the authors found several of my suggestions useful and will incorporate them in their paper. In addition, I appreciated their clarification on my more technical questions regarding Theorems 4 and 5.
>
> Currently, I maintain a positive view on this work recommending it for acceptance. However, I would refrain from increasing my score as I lack extensive knowledge of this area, and also in light of certain consistency results presented in this work being present in previous work (as pointed out by the other reviewers). I therefore maintain my original score.

---

### Official Review · Reviewer_9Sph · 2023-07-09

**Soundness:** 3 good
**Presentation:** 3 good
**Contribution:** 3 good
**Rating:** 7
**Confidence:** 2

**Summary:**

* The paper studies (Fisher, or “Bayes”) consistency in structured prediction.  In particular, the focus is on non-asymptotic, quantitative bounds for common and not-so-common (i.e., new) surrogate losses, which requires different proof techniques and an approach based on “H-consistency”.
* Thm 4, Thm 5 provide a few of the main results in the paper, showing that commonly used losses in structured prediction are not Bayes-consistent
* Thm 6, Thm 8 (and companion corollaries) provide the other main results in the paper, showing that a few new-ish but less tractable (though still convex) losses are H-consistent and Bayes-consistent — the losses in question here are the so-called “comp-sum” and “structured constrained” losses
* Finally, lem 10 shows that the latter (comp-sum) losses may be computed in polynomial time — but revealing something of a natural statistical/computational trade-off, i.e., the consistent losses evidently require more compute than the non-consistent ones

**Strengths:**

* The paper gives a quite detailed study on the statistical + computational aspects of H-consistency, for commonly (and not-so-commonly) used losses in structured prediction
* In particular, the paper highlights losses that are Bayes-consistent (though coming at the price of computability)


**Weaknesses:**

I have just a couple questions / comments:
* It seems the content of thms 4, 5 — on the lack of consistency — is also present in previous works (e.g., those by Ciliberto et al.).  Can you please elaborate on why these results in your paper are novel?
* Usually Fisher consistency is defined relative to a target loss function.  What is the target loss function here?
* I think the primary avenue the paper could be (significantly) strengthened is via an empirical analysis — the authors could illustrate the consistency (and computational costs) of the comp-sum, structured constrained losses through real-world examples.  That would really tie together the whole message of the paper.

**Questions:**

See above.

**Limitations:**

No they have not, but I’m not sure that’s necessary here.  This is an abstract / general theory paper.

---

> ### Author Rebuttal · Authors · 2023-08-08
>
> Thank you for your encouraging review. We will take your suggestions into account when preparing the final version. Below please find responses to specific questions.
>
> **1. It seems the content of thms 4, 5 — on the lack of consistency — is also present in previous works (e.g., those by Ciliberto et al.). Can you please elaborate on why these results in your paper are novel?**
>
> **Response:** Theorem 4 provides negative results for a broad and generalized family of loss functions, collectively referred to as structured max loss. This extends the scope of existing research, as previous works had only addressed the inconsistency of specific instances within the structured max loss category, such as Max-Margin Markov Networks (M3N) (e.g., studies by Osokin et al., Ciliberto et al. and Nowak et al.).
>
> Theorem 5 further elaborates on the negative results for voted conditional random field, a family of loss function that integrates the target loss $\ell(y’, y)$ within its formulation. To the best of our knowledge, no prior studies in the literature have explored the consistency of this specific formulation. The most closely related discussions center around a specialized instance of the multi-class logistic loss (also referred to as conditional random field in that context), in which $\ell(y’, y)$ disappears within the framework of the voted conditional random field. The previous work by Osokin et al., Ciliberto et al. and Nowak et al. point out that the multi-class logistic loss cannot be consistent in structured prediction due to the absence of the target loss function within its formulation. Instead, our result shows that, even when integrating the target loss $\ell(y’, y)$ within its formulation, the voted conditional random field cannot be consistent.
>
> **2. Usually Fisher consistency is defined relative to a target loss function. What is the target loss function here?**
>
> **Response:** The target loss function is the one described in lines 82 - 90, where $\ell$ is a symmetric loss defined over $\mathcal Y \times \mathcal Y$. For example, for sequences, $\ell$ may be the Hamming loss or some other rational loss. The surrogate losses typically adopted in structured prediction are expressed in terms of $\ell$. For example, for StructSVM, the surrogate loss is defined by $\mathsf L^{\text{StructSVM}} (h, x, y) = \max_{y' \neq y} \max \bigg\\{ 0, \ell(y', y) - (h(x, y) - h(x, y') ) \bigg\\}$.
>
> Our study of $H$-consistency bounds is general: we make no other assumption about $\ell$ beyond symmetry.
>
> **3. I think the primary avenue the paper could be (significantly) strengthened is via an empirical analysis — the authors could illustrate the consistency (and computational costs) of the comp-sum, structured constrained losses through real-world examples. That would really tie together the whole message of the paper.**
>
> **Response:** Thank you for the suggestion. Our paper mainly focuses on the theoretically principled surrogate losses for structured prediction based on $H$-consistency bounds. While we have demonstrated that the minimization of several proposed loss functions such as structured logistic loss benefit from efficient algorithms, we recognize the importance of further exploration. As such, we intend to dedicate future work to an extensive empirical analysis and the development of more universally applicable algorithmic solutions to encompass a broader family of surrogate loss functions as well as a diverse range of target losses.

---

> > ### Comment · Reviewer_9Sph · 2023-08-18
> > **Response to rebuttal**
> >
> > Thank you very much for the response.  I've gone through it and will maintain my score.

---

### Decision · Program_Chairs · 2023-09-21

**Decision:**

Accept (poster)

**Comment:**

This meta review is based on the reviews, the authors rebuttal and the discussions with the reviewers, discussions with the SAC, and ultimately my own judgement on the paper. There was a consensus that the paper contributes sound and interesting contributions to the field of structured prediction. I feel this work deserves to be featured at NeurIPS and will attract interest from the community. I would like to personally invite the authors to carefully revise their manuscript to take into account the remarks and suggestions made by reviewers. Congratulations!